

**Delayed Stormflow Generation in a Semi-humid Forested Watershed**
**Controlled by Soil Water Storage and Groundwater**
Zhen Cui, Fuqiang Tian[*]
Department of Hydraulic Engineering, State Key Laboratory of Hydroscience and Engineering,
Tsinghua University, Beijing 100084, China.
Corresponding author: Fuqiang Tian (tianfq@tsinghua.edu.cn)
**Key Points:**
• Delayed stormflow is initiated when soil water content reaches field capacity.
• Groundwater levels dictate the timing and magnitude of delayed stormflow.
• Rising groundwater levels enhance hydraulic conductivity and hydrologic connectivity,
driving delayed stormflow.



## Abstract

An analysis by Cui et al. (2024) of stormflow responses to rainfall in a mountainous forested watershed in the semi-humid regions of North China identified a distinct threshold for bimodal rainfall-runoff events, where delayed stormflow appeared to be influenced by shallow groundwater. This study further investigates the processes driving these bimodal events, focusing on the dynamics of soil water content (SWC) and groundwater level (GWL) during storm events. The results show that delayed stormflow is governed by the interplay between SWC and GWL. Delayed stormflow is initiated when SWC exceeds the soil's water storage capacity, while its timing and volume are determined by GWL fluctuations. During rainfall, SWC increases rapidly; if it does not reach the soil's water-holding capacity, it stabilizes after the rainfall ends. Conversely, if SWC surpasses the soil's storage capacity, it decreases rapidly post-rainfall, with the excess rainwater infiltrating deeper to recharge groundwater, leading to a gradual rise in GWL. As GWL rises, increased hydraulic conductivity facilitates the movement of shallow groundwater into the stream channel, resulting in delayed stormflow. Simultaneously, the effective connection area between the stream channel and adjacent hillslopes expands vertically. At specific high GWL thresholds, GWL responses across the watershed converge, significantly increasing groundwater discharge and reducing lag times, often causing the delayed stormflow peak to merge with the direct stormflow peak. These findings enhance our understanding of delayed stormflow generation in similar regions and contribute to refining runoff generation theories.

**Keywords:** Delayed stormflow; Soil water storage; Groundwater outflow; Stormflow generation mechanism; Hydraulic conductivity

## 1. Introduction

Previous research on stormflow processes in the Xitaizi Experimental Watershed (XEW) in North China identified a frequent occurrence of bimodal stormflow hydrographs, which often result in significant stormflow and associated flooding. Our findings highlighted that the onset of these



bimodal hydrographs is governed by threshold behavior, with delayed streamflow peaks emerging
when the combined total of event rainfall and antecedent soil moisture index exceeds 200 mm.
Additionally, we determined that delayed stormflow is primarily driven by contributions from shallow
groundwater. However, the mechanisms underlying the development of these bimodal hydrographs,
which represent complex emergent hydrological behaviors, remain poorly understood. Gaining
insight into the formation of delayed stormflow is crucial for advancing our understanding of
catchment runoff generation processes and improving flood forecasting capabilities.

Bimodal streamflow responses typically occur during the wetting-up phases of catchments.

Extensive research has examined the factors influencing dual streamflow peaks, revealing that
bimodal hydrographs are associated with threshold behavior linked to antecedent soil moisture,
antecedent precipitation, groundwater levels, soil water storage, and rainfall amount (Haga et al.,
2005; Graeff et al., 2009; Anderson and Burt, 1978; Padilla et al., 2015; Martínez-Carreras et al.,
2016). Despite these findings, the specific reasons for such threshold behavior—and how it leads to
the diverse shapes of stormflow hydrographs—remain inadequately explained. For instance,
Martínez-Carreras et al. (2016) documented a delayed peak only when watershed storage reached a
critical threshold of 113 mm. However, the precise reasons for this threshold and the processes that
follow are not fully understood.

The bimodal hydrograph is a significant manifestation of the nonlinear runoff response,

reflecting the complex interactions between runoff and rainfall. This nonlinear pattern provides
valuable insights into stormflow processes, including both the timing and magnitude of the response.
Recent decades have seen an increase in research on nonlinear and threshold changes in rainfall-
runoff responses, contributing to a deeper understanding of stormflow generation mechanisms.
Nonlinear shifts in stormflow, often characterized by rapid responses that can lead to flooding, have
been extensively documented (Detty and McGuire, 2010; Farrick and Branfireun, 2014; Graham et
al., 2010; Tromp-van Meerveld and McDonnell, 2006a; Penna et al., 2011; Scaife et al., 2020).
However, much of the existing literature lacks detailed exploration of the intricate mechanisms that
govern these shifts and the subsequent post-threshold runoff processes, leaving a gap in our



understanding of these complex hydrological dynamics across diverse catchments.
Bimodal runoff processes exemplify a typical nonlinear response of runoff, offering an intuitive
and effective way to simplify the description of complex hydrologic systems. Despite this, many
studies fail to distinguish between unimodal and bimodal streamflow responses, limiting our
understanding of these phenomena. Therefore, an in-depth investigation into the mechanisms driving
these responses is essential. Such research would enable the grouping of similar hydrologic responses
and facilitate comparisons of stormflow generation processes across different watersheds (Graham
and McDonnell, 2010; Tromp-van Meerveld and McDonnell, 2006a, b).
Observing substantial stormflow events in semi-humid regions is challenging due to the
relatively arid climate and lower runoff coefficients. Over the past decade, our analysis of 15 bimodal
stormflow events has provided valuable data and insights, contributing to the advancement of runoff
generation studies in similar regions. This study focuses on the dynamics of SWC and GWL to
investigate the processes underlying delayed stormflow patterns. The primary objectives are: (1) to
analyze the dynamics of SWC and GWL during storm events, (2) to elucidate the intrinsic
mechanisms driving the threshold behavior in bimodal hydrograph processes, and (3) to reveal the
generation mechanisms of delayed stormflow within the Xitaizi Experimental Watershed.

## 83    2. Materials and methods

### 84    2.1 Study site

The study was conducted in the Xitaizi Experimental Watershed (XEW), a small catchment in
North China, located approximately 70 km northeast of Beijing at coordinates 40°32′N and 116°37′E
(Fig. 1). XEW covers an area of 4.22 km², with elevations ranging from 676 to 1201 m above sea
level. The watershed is characterized by a monsoon-influenced semi-humid climate, with an average
annual precipitation of 625 mm, 80% of which occurs from June to September. The mean annual
temperature is 11.5°C, and the average relative humidity is 59.1%.

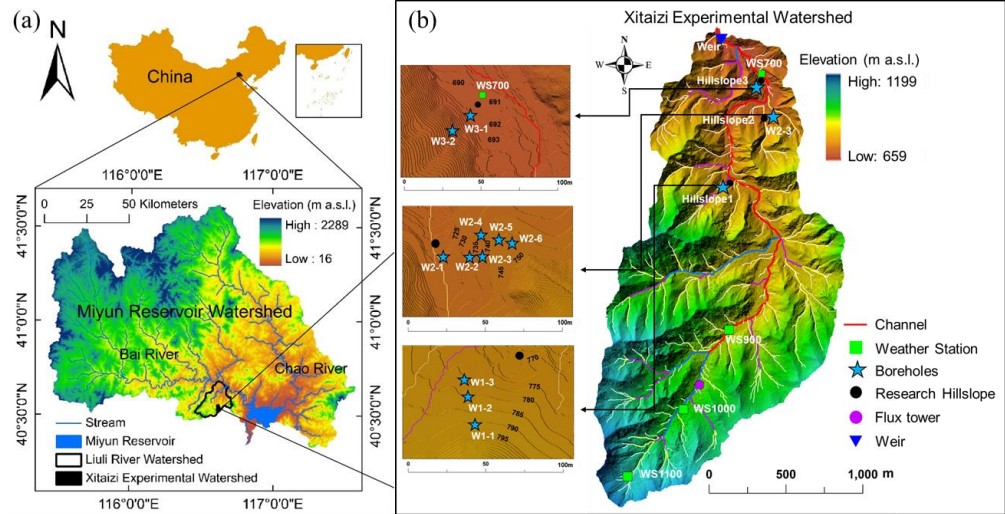

**Figure 1.** Location of Xitaizi Experimental Watershed (XEW), and the detailed distributed monitoring stations and instruments, including four weather stations, an outlet weir, eleven groundwater boreholes and five soil water profiles.

The geology of XEW is dominated by firmly compacted, deeply weathered granite, accounting for approximately 80% of the catchment area. The remaining bedrock consists of gneiss and dolomite. The watershed's soil types, primarily brown earth and cinnamon soil, extend to a depth of 1.5 meters with a saturated hydraulic conductivity ranging from 19.5 to 175.3 mm/h, averaging 45 mm/h. The catchment is heavily forested, with 98% coverage, comprising 54.2% broad-leaved, 2.3% coniferous, and 10.5% mixed forests. Shrubs cover the remaining 33% of the land.

**2.2 Meteorological and hydrological measurements**

Meteorological data were collected from 2013 to 2023 using four GRWS100 automatic weather stations, strategically distributed along an elevation gradient within XEW (Fig. 1). Air temperature and relative humidity were monitored using HC2S3-L probes, equipped with radiation shields, while photosynthetically active radiation was measured with LI-190R quantum sensors. Rainfall was recorded at 10-minute intervals using six tipping-bucket rain gauges located in open areas near the weather stations, with data averaged for analysis.



Streamflow was measured at the catchment outlet using a Parshall flume, with water levels
recorded every 5 minutes by a HOBO capacitance water level logger from 2014 onwards. The
recorded data were averaged to hourly intervals for analysis. Due to environmental challenges, some
data were lost, including stormflow data from July 19 to August 16, 2016, and streamflow data from
2018 to 2019, resulting in the exclusion of certain events from the analysis.

**2.3 Soil Water Content Observation**

Volumetric soil water content (SWC) was monitored using CS616 time-domain reflectometry
(TDR) probes at eight locations within the watershed. Data were recorded at 10-minute intervals,
with five sensors installed on Hillslope 1 and three near WS900 at 80 cm depth intervals. For this
study, the measurements were aggregated to hourly time steps, and the arithmetic mean SWC across
the profiles was used for analysis.

**2.4 Groundwater Level Observation**

Groundwater levels (below the ground surface, hereinafter referred to as bgs) were observed in
eleven 80 mm diameter boreholes distributed across three hillslopes within XEW (Fig. 1). Borehole
depths ranged from 5 to 26 m, penetrating weathered and fractured granite with varying degrees of
soil mantling. HOBO capacitance water level loggers recorded hourly groundwater levels. Boreholes
W1-1, W1-2, W2-4, W2-5, and W2-6 frequently registered no water levels, potentially due to
insufficient drilling depth. The saturated hydraulic conductivity of the weathered and fractured granite
was estimated to range from $5.2 \times 10^{-3}$ m/day to 1.16 m/day based on slug tests.
Groundwater levels were normalized using an index ($I_G$) calculated for each borehole following
the approach by Detty and McGuire (2010). The arithmetic mean of IG across all boreholes was used
to represent the overall groundwater level in the watershed.

**2.5 Separation of Rainfall-Runoff Events**

Rainfall events were identified using an intensity-based automatic algorithm described by Tian



et al. (2012). This algorithm defines event start and end times based on a threshold rainfall intensity
of >0.1 mm/h, with a minimum separation of six hours between events. Only events with cumulative
rainfall exceeding 5 mm were included in the analysis, resulting in the identification of 95 distinct
rainfall events from 2014 to 2023.
Storm runoff events were defined by a rapid streamflow increase and peak following rainfall.
Streamflow hydrographs were separated into baseflow and stormflow components using the HYSEP
program (Sloto & Crouse, 1996), with manual verification and adjustment based on straight-line
separation principles.
**2.3 Rainfall-runoff event analysis**
The analysis focused on understanding the conditions under which subsurface flow connects to
or disconnects from the stream. The dynamics among streamflow, SWC, and GWLs were examined
to reveal connectivity patterns, providing insights into the underlying processes. This simultaneous
observation of soil water, groundwater, and streamflow is defined as the soil water-groundwater–
stream response relationship.
Rainfall-runoff events, defined as those with total rainfall exceeding 5 mm and a corresponding
peak in streamflow, were analyzed. The peak rainfall intensity ($R_p$) was determined based on the
maximum 1-hour rainfall intensity, with the time of occurrence recorded as $TP_p$. As illustrated in Fig.
2, the initial streamflow ($Q_0$) was defined as the streamflow just before it began to rise, and $TQ_p$ was
the time when the maximum streamflow ($Q_p$) occurred. TRs and $TR_e$ represent the start and end times
of rainfall, respectively. The analysis of SWC and GWL dynamics followed a similar approach to
streamflow, replacing $Q_0$ and $TQ_p$ with $SWC_0$ ($I_{G0}$) and $TS_0$ ($TI_{G0}$), and $SWC_p$ ($I_{Gp}$) and $TS_p$ ($TI_{Gp}$),
respectively. This study analyzed 95 distinct rainfall-runoff events to better understand the
interactions between soil water, groundwater, and streamflow in response to rainfall.








**Figure 2.** Definition sketch for analysis of rainfall event.

## 3. Results

### 3.1. Relationship between SWC and GWL variability at the hillslope scale

We conducted a detailed analysis of the temporal evolution of SWC and GWL at the hillslope scale, focusing on their interactions across 95 rainfall-runoff events. Our analysis revealed a strong correlation between increases in GWL and elevated SWC values. During the early stages of these events, rainfall prompted a rapid rise in SWC, while GWL remained relatively stable. Once SWC reached a certain threshold, it either plateaued or gradually declined, coinciding with a marked increase in GWL. Subsequently, both SWC and GWL exhibited a nearly synchronous decline as GWL reached specific levels, manifesting in three distinct patterns of variation.

Figure 3 illustrates the response characteristics of SWC and GWL during three representative events. Red circles indicate periods of rainfall, while black circles represent post-rainfall periods. In a typical light rainfall event, as shown in Fig. 3a, the soil remained relatively dry with SWC values mostly below 0.20. During rainfall, SWC increased steadily until the rainfall ceased, after which it stabilized, while GWL showed minimal change.



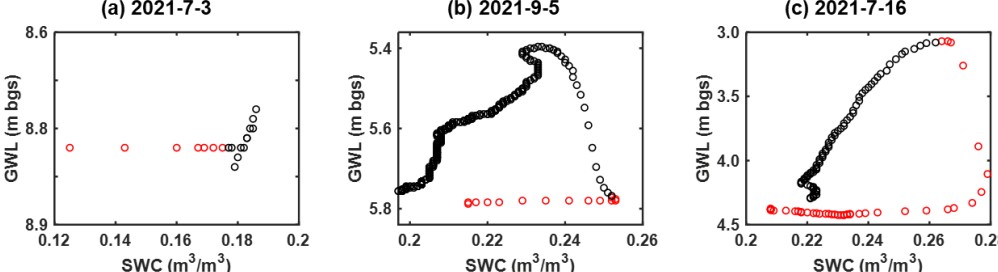

**Figure 3.** Three typical SWC-GWL dynamics patterns during rainfall-runoff events.

Figures 3b and 3c depict the dynamics of SWC and GWL during storm events, where a pronounced counterclockwise hysteretic relationship was observed. Both SWC and GWL exhibited significant increases, with SWC surpassing 0.20. The primary distinction between these patterns lies in the timing of the GWL rise: in Fig. 3b, GWL began to rise after the rainfall ended, whereas in Fig.3c, GWL started to rise noticeably before the end of the rainfall.

In the scenario represented by Fig. 3b, SWC continued to rise during rainfall while GWL remained largely unchanged. After the rainfall ceased, SWC began to decline, and GWL subsequently rose before eventually falling. Fig. 3c, which typically represents extreme storm events, shows that when SWC exceeded 0.25, GWL rose sharply as SWC continued to increase until it reached 0.28. Despite ongoing rainfall, SWC then decreased, and GWL experienced a significant surge, continuing until the rainfall stopped, after which both variables began to decline. This pattern underscores the complex dynamics of SWC and GWL during storm events, highlighting the nuanced responses of these hydrological parameters.

We further quantified the frequency of SWC and GWL increases or decreases across the 95 rainfall-runoff events. As depicted in Fig. 4, there were 49 events where SWC increased and 43 events where GWL increased. Among these, 26 events saw a decline in SWC, and 15 experienced a decline in GWL, with only 15 events showing a decrease in both variables. These 15 events were associated with delayed stormflow and generated larger stormflow volumes. Notably, the rainfall-runoff event on August 15, 2021, featured a more dispersed rainfall distribution, with multiple fluctuations in SWC and GWL throughout the event. Consequently, our subsequent analysis primarily focused on the





remaining 14 events.

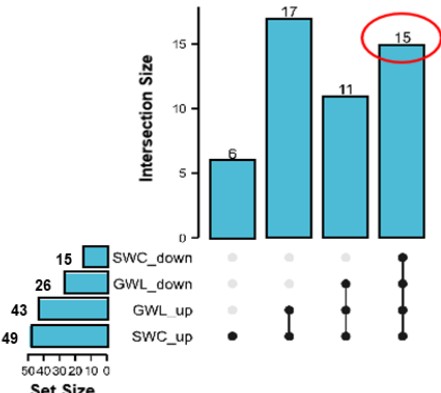


**Figure 4**. Upset plot of soil water content and groundwater level response characteristics.
**3.2 Soil water content dynamics during storm events**

Figure 5 presents the SWC dynamics observed during 14 distinct rainfall-runoff events, each

characterized by minimal or no intermittent rainfall during the recession period. To facilitate a clear
comparison of SWC changes across different events, the peaks of all events were aligned to the
position corresponding to a horizontal axis value of 0.

During the initial phases of these events, rainfall triggered a rapid increase in SWC, which

quickly reached its peak. In the recession phase, the rate of SWC decline slowed as SWC decreased,
eventually stabilizing around 0.20. This pattern of SWC variation is schematically represented in Fig.

6.





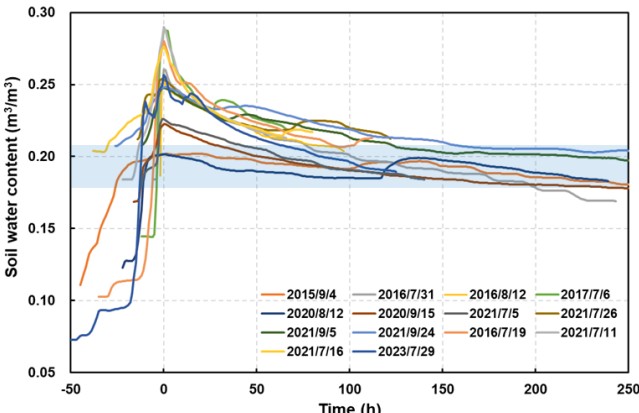


**Figure 5.** Soil water content dynamics during various storm events.

The SWC response to rainfall was found to be extremely rapid. Upon the onset of rainfall, SWC
quickly increased. Once the rainfall ceased, the subsequent behavior of SWC was dependent on its
peak value. If SWC was less than or equal to 0.20, it either stabilized for a period or decreased very
slowly. However, if SWC exceeded 0.20, it decreased rapidly, eventually stabilizing around 0.20. The
presence of a peak in SWC was determined by whether it surpassed the 0.20 threshold; the greater
the excess above 0.20, the more rapid the subsequent decline.

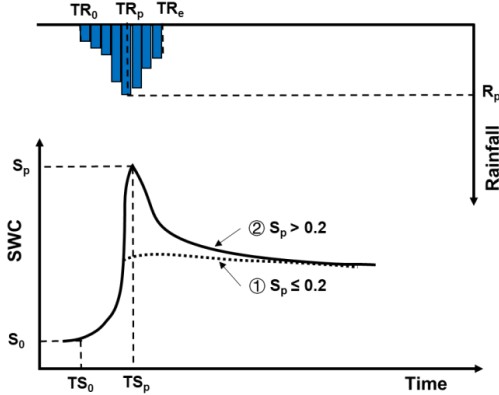



**Figure 6.** Schematic diagram of soil water content response process during storm events. $S_p$ is the maximum value of soil water content.

### 3.3 GWL dynamics during storm events

During storm events, we identified two distinct types of GWL responses: quick and slow. These response types are conceptually illustrated in Fig. 7. The GWL with a quick response typically exhibits a distinct process curve compared to the slower response. The quick GWL response is closely aligned with a swift increase in soil water content (SWC), lagging the SWC peak by just 0 to 6 h (Fig. 7a). In scenarios where SWC exceeds 0.20, particularly beyond 0.24, the GWL often shows a secondary increase following its initial peak, marked by the dotted line in Fig. 7a. Conversely, a slower GWL response, depicted in Fig. 7b, occurs when SWC declines sharply after peaking.

Analysis of GWL variations across hillslope positions revealed that GWL in HS2 (W21-23) exhibits the rapid response type (Fig. 7a), while GWL in HS1 (W13) and HS3 (W31 and W32) demonstrates the slower response type (Fig. 7b). These response patterns suggest that the GWL dynamics are not only influenced by SWC but are also dependent on the specific hillslope's geological structure.

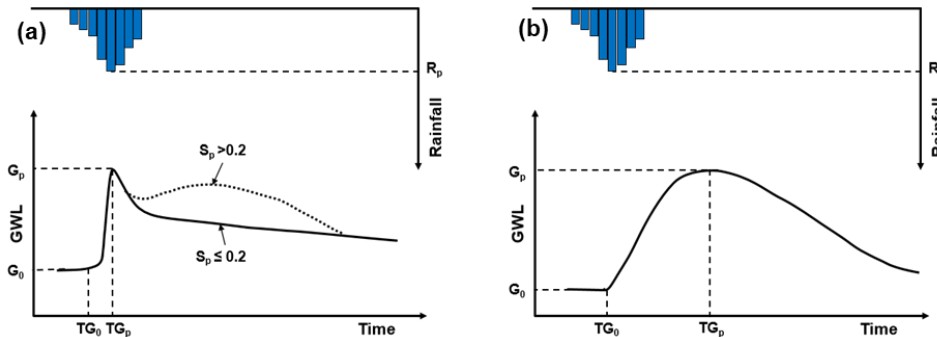

**Figure 7.** Schematic diagram of groundwater level response during storm events. $G_0$ and $G_p$ are the initial and maximum values of groundwater level respectively. $S_p$ is the maximum value of soil water content.

Further examination of GWL responses at various locations is presented in Fig. 8, which details the magnitude of GWL increases and the lag times relative to rainfall onset for each event. Despite



variability in GWL across observation wells, with the exception of W21 and W31 (which are located
at the foot of the hillslope and exhibit smaller GWL changes), the differences in GWL increments at
other wells are relatively minor, with mean increases ranging from 1 to 2 meters. On the same
hillslope, GWL increments generally increase progressively from the foot to the top (e.g., W21, W22,
and W23 on HS2, and W31 and W32 on HS3).

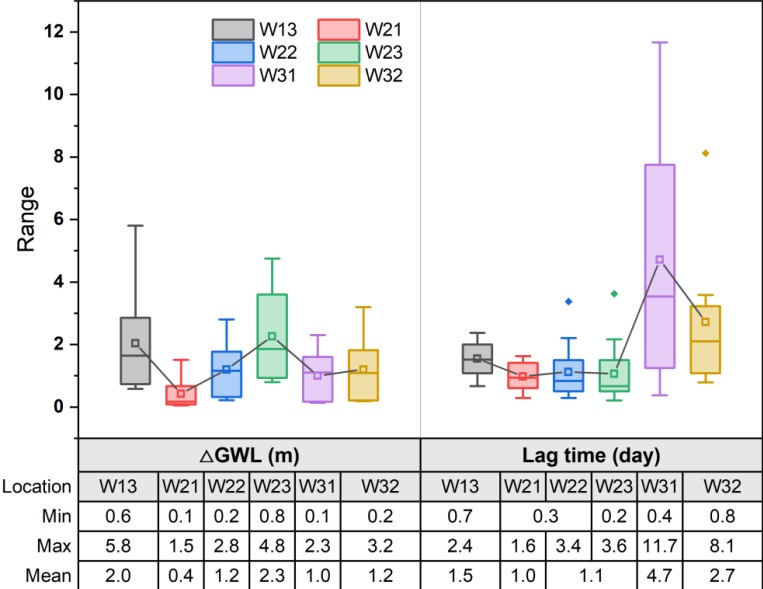

| Location | ΔGWL (m) | | | | | | Lag time (day) | | | | | |
|---|---|---|---|---|---|---|---|---|---|---|---|---|
| | W13 | W21 | W22 | W23 | W31 | W32 | W13 | W21 | W22 | W23 | W31 | W32 |
| Min | 0.6 | 0.1 | 0.2 | 0.8 | 0.1 | 0.2 | 0.7 | 0.3 | | 0.2 | 0.4 | 0.8 |
| Max | 5.8 | 1.5 | 2.8 | 4.8 | 2.3 | 3.2 | 2.4 | 1.6 | 3.4 | 3.6 | 11.7 | 8.1 |
| Mean | 2.0 | 0.4 | 1.2 | 2.3 | 1.0 | 1.2 | 1.5 | 1.0 | | 1.1 | 4.7 | 2.7 |


**Figure 8.** Groundwater level increments (ΔGWL) and lag time of peak water level relative to
rainfall onset across locations.
However, lag times for reaching maximum GWL exhibit greater variation across locations. For
instance, at HS3, the delay for maximum GWL in W31 ranged from 0.4 to 11.7 days, and in W32
from 0.8 to 8.1 days, both of which are longer than the lag times observed at HS1 (0.7 to 2.4 days)
and HS2 (0.2 to 3.6 days). There is no clear correlation between the lag time of maximum GWL and
its distance from the hillslope foot within the same hillslope. These discrepancies in lag times between
different hillslopes may be attributed to spatial variations in geological conditions, as suggested by
Kosugi et al. (2011) and Padilla et al. (2015).





### 3.4 Characterization of GWL response on different hillslopes

Figure 8 reveals that while the magnitude of GWL increments across various locations remains relatively consistent, the lag time for GWL to reach its maximum value exhibits substantial variation. To further investigate these dynamics, we analyzed the relationship between GWL increments and SWC across 14 storm events. In Fig. 9, the length of the orange bar represents the GWL increment during the phase when SWC increased to its peak, while the green bar indicates the GWL increment during the phase when SWC decreased from its peak until GWL reached its maximum. The black and red dotted lines mark the initial SWC and the SWC at the onset of GWL rise, respectively. Locations without bars in Fig. 9b, e, and f indicate missing data.

The analysis shows that a significant increase in SWC from its initial value following rainfall is indicative of a delayed GWL response. Specifically, the larger the difference between the SWC at the onset of GWL rise ($SWC_G$) and the initial SWC ($SWC_0$), the later the GWL rise begins. Conversely, if $SWC_G$ and $SWC_0$ are closely aligned, the GWL begins to rise almost simultaneously with the increase in SWC.

For example, at HS1 (W13), GWL began to rise only after SWC exceeded 0.20, and the majority of the GWL increase occurred during the SWC decline phase. This suggests that the GWL response on HS1 is influenced by soil wetness, indicating a potential threshold effect of SWC on GWL dynamics. On HS2 (including W21-23), the GWL response was more immediate, with increases closely following SWC rises. Here, the SWC at the onset of GWL rise varied widely, ranging from 0.13 to 0.26, and was generally close to the initial SWC, suggesting that GWL increases at these locations are less dependent on SWC thresholds. HS3 exhibited both quick and slow GWL responses. The initial response occurred soon after the SWC increase but at a slow rate that persisted over an extended period. The majority of the GWL increment at HS3 occurred during the SWC decline phase after its peak.

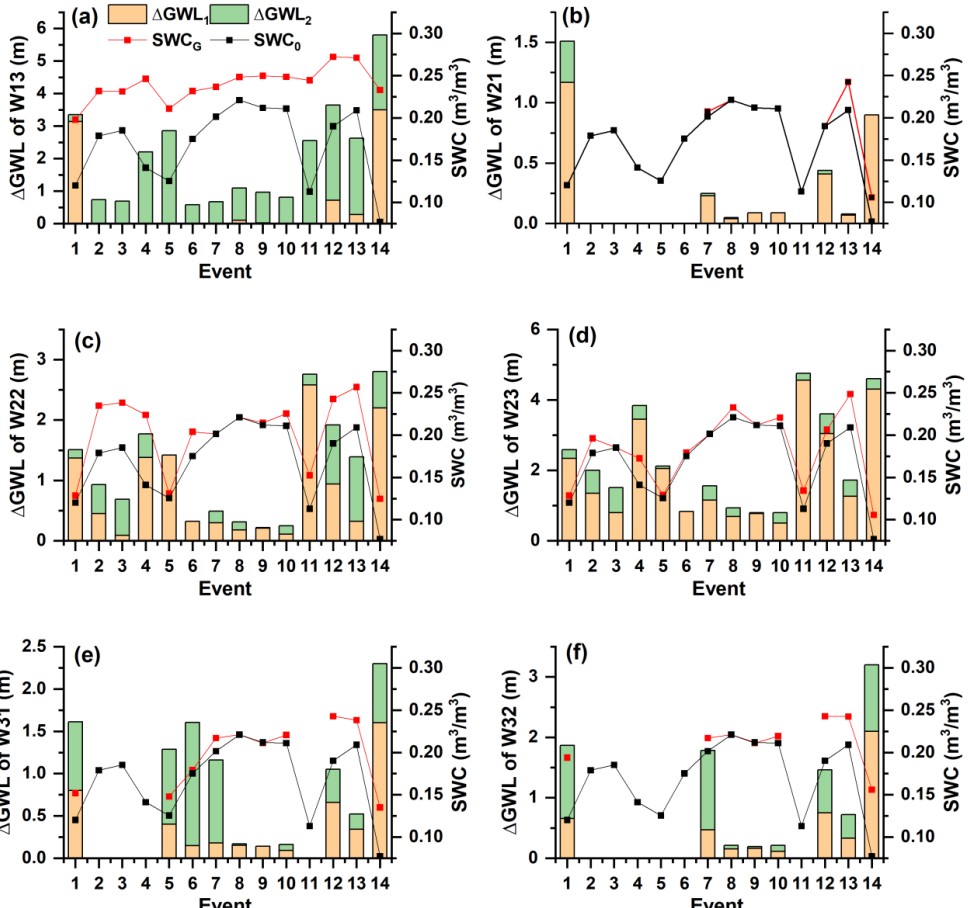

**Figure 9.** Groundwater level increments (ΔGWL) across various locations during 14 storm events, along with initial soil water content (SWC$_0$) and corresponding soil water content at groundwater level initiation (SWC$_G$).

These findings indicate that the emergence of quick and slow GWL response types is strongly linked to SWC dynamics. In quick response types, GWL growth primarily occurs during the SWC increase phase, resulting in a steep response curve. In slow response types, GWL growth predominantly occurs during the SWC decline phase after the peak, leading to an arch-shaped response curve. This distinction underscores the critical role of SWC dynamics in governing the timing and magnitude of GWL responses across different hillslopes.



**3.5 Inter-hillslope dynamics in GWL response**

To explore the differences in GWL response times across hillslopes, particularly the delayed occurrence of maximum GWL on HS3 compared to HS1 and HS2, we quantified the lag times of GWL responses on HS1 ($t_{S1}$), HS2 ($t_{S2}$), and HS3 ($t_{S3}$) relative to the onset of rainfall. We then calculated the elapsed time differences between $t_{S1}$ and $t_{S3}$, as well as between $t_{S2}$ and $t_{S3}$ ($\Delta t = t_S - t_{S3}$) and analyzed their correlation with peak $I_G$, as shown in Fig. 10. Negative $\Delta t$ values indicate that the GWL peaks on HS1 and HS2 occurred earlier than on HS3.

A significant negative linear relationship was observed between $\Delta t$ and peak $I_G$, described by the equation $I_G = -0.0015 \times \Delta t + 0.27$ ($R^2 = 0.78$, $p < 0.001$). As GWL increased , eventually approaching zero when peak $I_G$ exceeded 0.30, with only minor fluctuations, particularly during extreme storm events. Notably, although Fig. 10 labels the vertical axis as $I_G$ to represent watershed GWL status, a similar pattern emerges when replacing $c$ with GWL at any specific location, though the GWL thresholds vary across different sites.

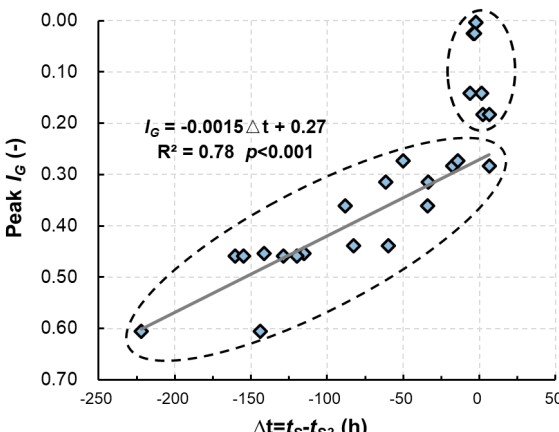

**Figure 10.** Correlation between peak $I_G$ and the elapsed times from $t_{S1}$, $t_{S2}$ to $t_{S3}$ ($\Delta t = t_S - t_{S3}$). $t_{S1}$, $t_{S2}$ and $t_{S3}$ are the lag time of peak groundwater levels on HS1, HS2 and HS3, respectively.

These results suggest that as GWL increases, the GWL response across different hillslopes tends to synchronize. This synchronization may be attributed to the enhanced hydraulic conductivity as



GWL rises, as noted by Padilla et al. (2015), who reported that shorter lag times in bedrock
groundwater are associated with high-transmissivity conduits. When a critical GWL threshold is
reached, the water transport capacity increases substantially, leading to nearly simultaneous responses
across hillslopes. This finding is consistent with Scaife et al. (2020), who observed increased
hydrological connectivity between hillslopes and the stream channel during such conditions.

In summary, these results indicate that as GWL rises, not only does groundwater recharge more

rapidly from infiltrating rainfall, but it also reaches the stream channel more quickly through more
transmissive layers, which can be explained by the transmissivity feedback mechanism (Kendall et
al., 1999; Bishop et al., 2011).
**4. Discussion**
**4.1 Characterization of groundwater response at the watershed scale**

The groundwater level (GWL) response to storm events exhibited spatial variability across the

watershed. $I_G$, which represents the average normalized GWL across different locations, provides a
comprehensive view of the watershed's GWL dynamics. Our analysis of $I_G$ revealed that, compared
to individual well GWL changes, $I_G$ often exhibits two distinct peaks during storm events. Specifically,
among the 14 events analyzed, 9 events showed two $I_G$ peaks, which coincided with the occurrence
of two streamflow peaks. In contrast, only wells W13 and W23 exhibited dual GWL peaks: W13
displayed two peaks during one event, while W23 exhibited two peaks in five events, with the
remaining wells showing only a single peak (see Table 1).
**Table 1.** Statistical results of response characterization of streamflow, $I_G$ and groundwater levels.

|  |  |  | HS1 |  | HS2 |  | HS3 |  |
|---|---|---|---|---|---|---|---|---|
|  | **Streamflow** | $I_G$ | **W13** | **W21** | **W22** | **W23** | **W31** | **W32** |
| Total number of events | 14 | 14 | 14 | 8 | 14 | 14 | 9 | 9 |
| Number of events with two peaks | 9 | 9 | 1 | 0 | 0 | 5 | 0 | 0 |





Figure 11 illustrates the timing of $I_G$ peaks relative to the soil water content (SWC) response.
The first $I_G$ peak occurred rapidly following rainfall, initiating 0-2 h after the SWC began to rise and
peaking 0-9 h later (average 3.7 h) after the SWC reached its maximum. The second $I_G$ peak typically
occurred post-rainfall, lagging behind the SWC peak by 10-65 h (average 28 h). These response
patterns are consistent with the quick and slow GWL response types identified in section 3.2. The
occurrence of two $I_G$ peaks is primarily attributed to the superimposition of groundwater contributions
from different hillslopes, each with distinct response rates. The fast GWL response is closely linked
to immediate rainfall and rising SWC, whereas the slow GWL response occurs over a broader
timescale, emphasizing the need for further attention to the latter in hydrological studies.

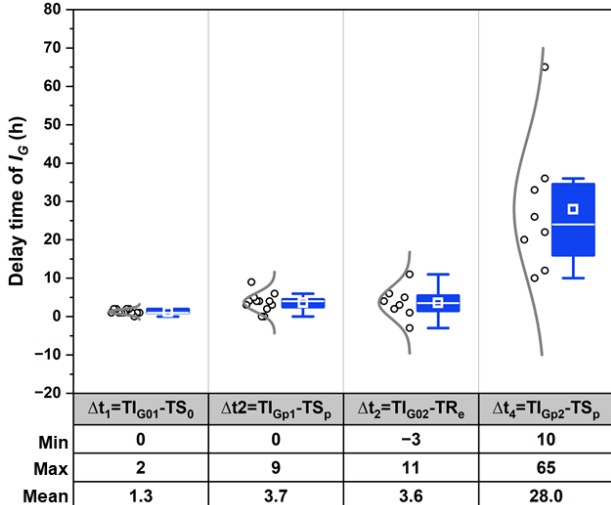


**Figure 11.** Delay time of $I_G$ peaks relative to peak soil water content. $TI_{G01}$ and $TI_{G02}$ are the onset
times of the first and second peaks of $I_G$, respectively. $TS_0$ and $TS_p$ are the time when soil water
content started to increase and peaked, respectively. $TI_{Gp1}$ and $TI_{Gp2}$ are the time when $I_G$ started to
increase and peaked, respectively. $TR_e$ is when the rainfall ends.

The growth rates of $I_G$ towards the two peaks in various events were quantified (Fig. 12). A
notable disparity was observed between the growth rates of the two peaks, with the first peak ($r_1$)
displaying a significantly higher rate than the second peak ($r_2$). Specifically, $r_1$ ranged from 0.03 to



0.98 per day (average 0.38/day), while $r_2$ ranged from 0.01 to 0.31 per day (average 0.07/day). These
two peaks correspond to the quick and slow GWL responses across different hillslopes. In events
featuring two $I_G$ peaks, the maximum $I_G$ typically occurred at the second peak. Additionally, in events
characterized by higher GWLs (lower $I_G$), the difference between the growth rates of the two $I_G$ peaks
diminished, making them more difficult to distinguish, as observed in events 9 and 10. In events 11-
14, which had much higher GWLs, only a single peak was observed in the $I_G$ process. This outcome
aligns with the findings presented in Fig. 10, indicating that higher GWLs lead to a more synchronized
GWL response across the watershed.

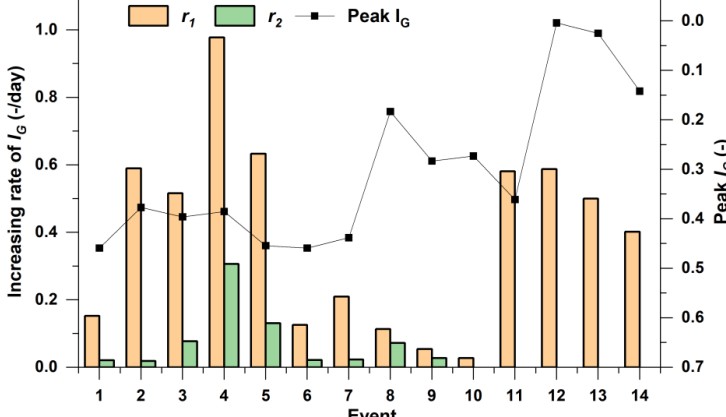


**Figure 12.** Growth rates of $I_G$ and its maximum value across various events. $r_1$ and $r_2$ denote the
rates of ascent during the periods when $I_G$ reaches its first and second peaks, respectively.

The first $I_G$ peak, which occurred rapidly during rainfall and was closely associated with rising

SWC, contrasts with the second $I_G$ peak, which appeared post-rainfall when the soil layer began
draining. The formation processes and underlying mechanisms of these two $I_G$ peaks are distinct. As
Dang et al. (2023) noted, rainfall generates pressure waves that rapidly expel soil water from the
column's bottom, while infiltrated rainwater migrates slowly downward. This change in head within
the surface soil layer can induce an immediate GWL response. We hypothesize that the fast $I_G$ peak
may result from increased SWC inducing a kinematic wave, which displaces "old" soil water and
groundwater ahead, leading to a near-synchronous GWL rise (e.g., Anderson and Burt, 1978).



Although water flow through soil and bedrock is slow, the theoretical celerity of this response is
instantaneous, hence the rapid GWL rise. Furthermore, early drilling data revealed the presence of
faults in the rock structure of HS2, which may contribute to the faster GWL response on this hillslope
compared to others. The slow $I_G$ peak likely forms as rainwater gradually infiltrates through the soil
and bedrock layers, eventually recharging the groundwater. Crucially, there exists a threshold for the
soil layer's water storage capacity: before SWC reaches this threshold, all rainfall is retained within
the soil layer. Once this critical threshold is surpassed, the soil layer cannot retain additional water
for extended periods and swiftly releases excess water to deeper layers, leading to a subsequent
reduction in SWC while the GWL rises due to effective groundwater replenishment from infiltrated
rainwater.
**4.2 Delayed stormflow processes dynamically aligned with GWL dynamics**
Previous studies (Cui et al., 2024) indicated that during heavy rainfall, the streamflow in the
XEW exhibits a bimodal hydrograph, with delayed stormflow likely formed by shallow groundwater
outflow. Assessing the relative timing and lags between groundwater and streamflow responses is
crucial for understanding dominant runoff generation processes (Beiter et al., 2020). Inconsistencies
in response times may indicate the contribution of alternative water sources to the stream channel.
Fig. 13 illustrates the timing of maximum $I_G$ ($I_{Gp}$) and maximum SWC ($SWC_p$) responses for eight
storm events, as well as the rainfall duration. Each horizontal bar represents the onset of rain on the
left end and the lag time for the maximum value on the right end of the corresponding variable, except
that the bar length for $t_{Rain}$ indicates the duration of the rainfall.



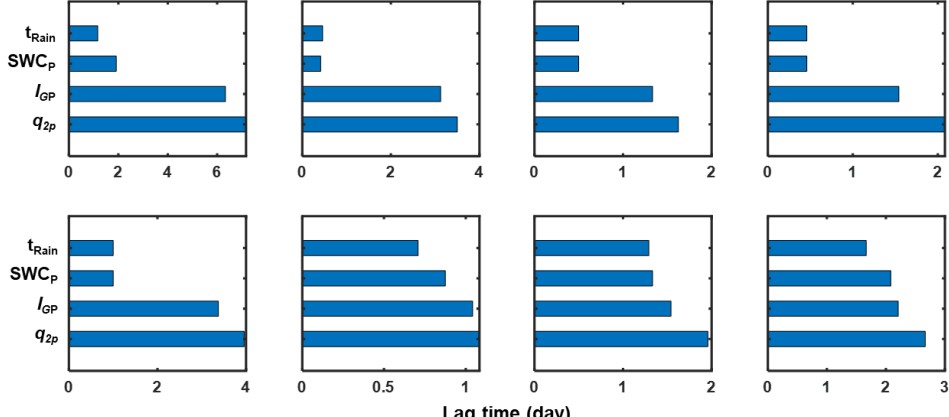


**Figure 13.** Lag times of maximum soil water content and groundwater level relative to rainfall onset. The beginning and end of each colored bar indicate the rise and peak times of according variable. $SWC_{SP}$ is maximum soil water content. $I_{GP}$ is the maximum $I_G$ and $q_{2p}$ is the delayed streamflow peak.

Rainfall duration across events ranged from 0.46 to 1.67 days. SWC, $I_G$, and delayed stormflow ($q_{2p}$) successively reached their peak values following the onset of a storm. SWC responded rapidly to rainfall, with its peak occurring 0.4 to 2.1 days after the storm began, typically coinciding with or slightly after the end of the rainfall. $I_G$ continued to increase after the SWC peak, reaching its maximum before the peak in $q_{2p}$. The lag times from the $SWC_p$ to the $I_{Gp}$ and $q_{2p}$ varied considerably across events. However, the lag time between $I_{Gp}$ and $q_{2p}$ was relatively consistent across events. As the prior research of Haught and Meerveld (2011) and Rinderer et al. (2016), who reported that identical or earlier response timing of groundwater compared to streamflow implies that a robust hillslope-stream connectivity is established and the streamflow response is driven by hillslope groundwater. Our results reinforce this understanding, as the timing of $q_{2p}$ was predominantly governed by changes in groundwater. This conclusion is further corroborated by the strong linear correlation between the lag times of $q_{2p}$ ($t_{2p}$) and $I_{Gp}$ ($t_{IGp}$), as indicated by the regression equation $t_{2p}$ = 1.11× $t_{IGp}$ + 0.17, with a slope of 1.11, a determination coefficient $R^2$ = 0.995, and a t-test significance at the 0.01 level. (Fig. 14). Conversely, the correlation between $t_{2p}$ and $SWC_p$ ($t_{SWCp}$) was weak, with $R^2$ = 0.029 and a t-test significance level of 0.688, well above the 0.05 threshold,



indicating that $t_{SWCp}$ has a negligible impact on $t_{2p}$. Additionally, as shown in Fig. A1, the $I_G$ process
lines during the delayed stormflow period closely mirrored the shape of the streamflow hydrograph,
further emphasizing the dominant role of $I_G$ in controlling $q_{2p}$.

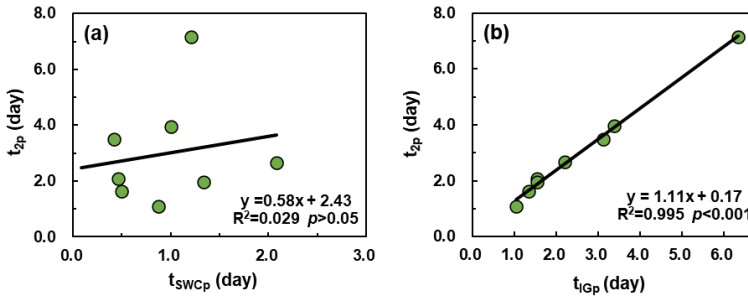


**Figure 14.** Lag times of maximum (a) SWC and (b) $I_G$ in relation to the lag times of delayed
streamflow peaks ($t_{2p}$). $t_{SWCp}$ and $t_{IGp}$ are the occurrence times of the maximum soil water content
and maximum $I_G$, respectively.

The delayed stormflow process was quantitatively analyzed in relation to $I_G$ variations during

this phase. As shown in Fig. 15, during the delayed stormflow period (i.e., non-rainfall period) of the
eight bimodal events, streamflow demonstrated a strong exponential relationship with increases in
GWL ($IG_p$), with a highly significant correlation ($p < 0.001$) and a correlation coefficient of $R^2 = 0.90$.
This exponential rise in streamflow corresponding to the increase in GWL can be attributed to a
potential enhancement in lateral hydraulic conductivity as the water table approaches the land surface,
consistent with findings by Detty and McGuire (2010) and Kendall et al. (1999). Furthermore, the
rapid increase in streamflow as the water table enters the surficial zone flattens the GWL vs.
streamflow curve, indicating the occurrence of transmissivity feedback. This feedback mechanism
led to a rise in GWL, which mobilized groundwater outflow, facilitating rapid transport via shallow
flow paths to the stream, as described by Lundin (1982).



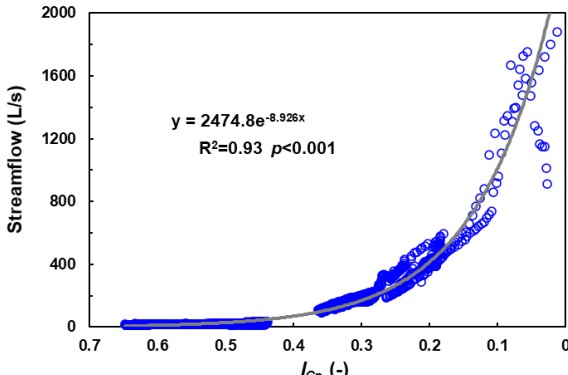


**Figure 15.** Correlation between $I_G$ and streamflow during delayed stormflow periods.

**4.3 Initiation of delayed stormflow triggered by soil water storage**

Understanding the activation thresholds that govern water movement is crucial, as emphasized

by McDonnell et al. (2021). Previous analyses in this study identified a strong correlation between
delayed stormflow and the slow response of GWL. This slow response is triggered by a rapid decline
in SWC, which only occurs when SWC exceeds a critical threshold of 0.20. To pinpoint the control
threshold for delayed stormflow in XEW, we analyzed 63 out of 95 rainfall-runoff events that had
complete SWC and GWL data. The relationship between SWCp and $q_s$ for these events is illustrated
in Fig. 16. A distinct threshold phenomenon was observed: $q_s$ remained minimal when SWC was
below 0.20, a condition prevalent in nearly all unimodal events. However, when SWC surpassed 0.20,
there was a sudden increase in $q_s$ due to the emergence of delayed stormflow in some events. Notably,
when SWC exceeded 0.23, a significant surge in stormflow volume occurred, with a second
stormflow peak appearing in all events. This suggests that an SWC range of 0.20 to 0.23 reflects the
soil layer's water storage capacity, serving as a critical threshold for the onset of delayed stormflow.



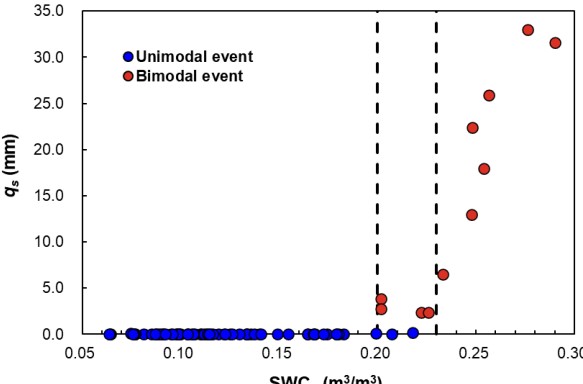


**Figure 16.** Relationship between maximum soil water content (SWC$_p$) and event stormflow amount
($q_s$).
These findings highlight the pivotal role of the surface soil layer's water deficit or water-holding
capacity in determining the rainfall threshold for delayed stormflow initiation. During rainfall events,
rainwater was largely retained within the soil layer until the amount exceeded its water-holding
capacity. Furthermore, the analysis revealed that despite fluctuations in SWC, the $q_s$ generated during
unimodal events consistently remained below 1 mm, indicating that stormflow in these cases was
mainly due to direct rainfall interception by the channel. Given the varying soil layer depths across
the watershed, more detailed data on soil depth and distribution are essential for accurately estimating
watershed-wide soil water storage capacity. However, observations of SWC across different locations
show minimal variability within the watershed, suggesting that SWC serves as a reliable indicator of
soil water storage in this study.
**4.4 Conceptual model of runoff generation mechanism in XEW**
In this section, we present a conceptual model that elucidates the mechanisms of runoff
generation in the XEW watershed, with a focus on the role of soil water storage and GWL dynamics.
Soil water storage is identified as the critical factor driving the transition from initial to delayed runoff
generation. Once the soil water deficit is replenished, the slowly rising GWL becomes the primary
control on the delayed stormflow process. Fig 17 illustrates the conceptual model of runoff generation,
which incorporates transmissivity feedback mechanisms to explain the formation of distinct





hydrographs.

**1.  Runoff generation under dry conditions (Fig. 17b):**

When the watershed is relatively dry and experiences light rainfall, the model shows that rainwater primarily infiltrates and is stored within the soil layer. During such events, the streamflow is composed of two main components: (1) a rapid streamflow peak resulting from direct rainfall on the channel and (2) baseflow originating from the gradual release of deep groundwater. This baseflow contribution is relatively constant, reflecting the slow discharge of groundwater from deeper aquifers.

**2.  Delayed stormflow during moderate storms (Fig. 17c):**

In more substantial storm events, the soil water storage capacity is exceeded after the soil water deficit is fully replenished. The initial response is similar to the dry condition scenario, with a rapid streamflow peak generated by direct rainfall on the channel. However, as rainfall continues, the excess water infiltrates deeply, elevating the GWL and expanding the effective connection area between the stream channel and adjacent hillslopes. This vertical expansion of the saturated zone allows a significant volume of shallow groundwater to be rapidly conveyed to the channel as the GWL reaches a more conductive soil layer. The result is a delayed stormflow peak, which occurs after the rainfall has ended.

**3.  Runoff generation during extreme storm events (Fig. 17d):**

In extreme storm events characterized by substantial rainfall input, GWLs rise sharply across the entire watershed, reaching levels associated with higher hydraulic conductivity. This synchronous rise in GWL triggers the rapid release of a large volume of shallow groundwater, leading to a significant flood peak within a short time frame. During these events, the GWL in the riparian zone may rise into the soil layer or even reach the ground surface, facilitating water movement into the channel via soil subsurface flow. Observations from extreme storm events support this mechanism, as the deeper soil layer in the riparian zones often shows a sudden and sustained increase in SWC after rainfall has ceased, suggesting that groundwater from adjacent hillslopes may be replenishing soil water in these areas.



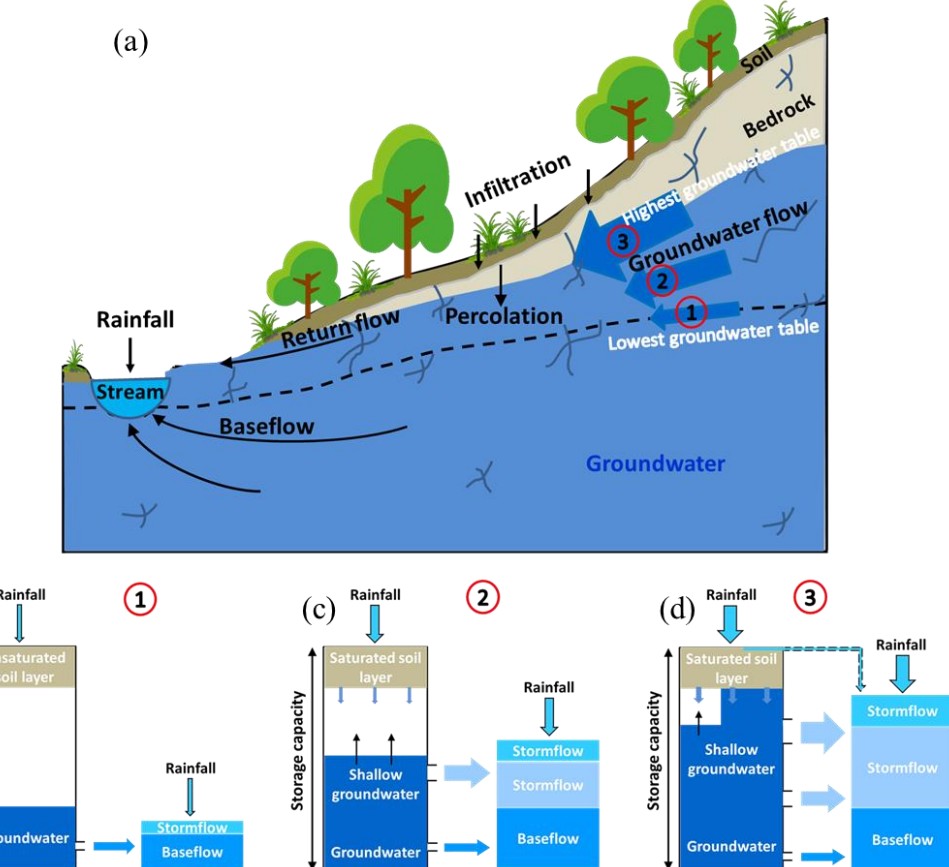

**Figure 17.** Conceptual model illustrating the stormflow generation associated with the transmissivity feedback.

The transition from the runoff generation model depicted in Fig. 17b to Fig. 17c and ultimately Fig. 17d corresponds to a progressive wetting-up of the watershed. The abrupt changes in stormflow volume and timing are initially triggered by soil water storage and later governed by the hydraulic conductivity of the bedrock and micro-topography. This conceptual model provides a quantitative framework for understanding how varying hydrological conditions influence the runoff generation processes in the XEW watershed.



## 5. Conclusions

Building upon previous work that identified and characterized bimodal streamflow patterns in the XEW watershed, this study provides a detailed, quantitative analysis of SWC and GWL at the event scale to elucidate the mechanisms behind delayed stormflow generation. The findings reveal that when soil water storage surpasses its holding capacity, a secondary increase in streamflow is triggered. This secondary, or delayed, stormflow is primarily governed by GWL dynamics, influencing both the magnitude of delayed stormflow and the lag time to its peak.

During rainfall events, SWC exhibits a rapid response, continuing to increase until it reaches or exceeds the soil's water storage capacity. If the stored water remains within this capacity, SWC stabilizes or decreases slowly after rainfall ceases, eventually leveling off near the field capacity. The rate of SWC decrease is directly proportional to the extent to which it exceeds field capacity. When SWC decreases, excess rainwater infiltrates deeper into the soil, raising the GWL. Once the GWL begins to rise, it becomes the dominant factor driving the delayed stormflow process.

As GWL rises, hydraulic conductivity increases, allowing more groundwater to flow from hillslopes into the channel, thereby forming delayed stormflow. This process also causes the effective connection area between the stream channel and adjacent hillslopes to expand vertically. At specific high GWL thresholds, the synchronization of GWL responses across multiple hillslopes leads to a substantial increase in stormflow volume. This synchronized response shortens the lag time and increases the volume of delayed stormflow, often causing the delayed stormflow peak to merge with the direct stormflow peak.

These findings provide a deeper understanding of the nonlinear behavior of stormflow and the mechanisms behind the formation of bimodal hydrographs. This enhanced understanding has significant implications for advancing hydrological theory and offers valuable insights for improving and optimizing flood modeling and prediction.



**Data availability**

All the data used in this study will be available at the Zenodo website at the time of publication.

**Author contribution**

ZC contributed the conceptualization, formal analysis, investigation and writing; FT contributed the conceptualization, formal analysis and revision.

**Competing interests**

Fuqiang Tian is a members of the editorial board of Hydrology and Earth System Sciences.

**Acknowledgements**

This study was supported by National Key R&D Program of China (2022YFC3002902) and National Natural Science Foundation of China (51825902).

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





**Appendix A:**



**Figure A1.** Examples of responses of streamflow, IG and soil water content to rainfall.