# Peer review of "Delayed Stormflow Generation in a Semi-humid Forested Watershed"

_EGUsphere, 2024_

## Referee Comment (RC1)

Review of manuscript EGUSPHERE-2024-2177. Delayed Stormflow Generation in a Semi-humid Forested Watershed Controlled by Soil Water Storage and Groundwater.

**Specific comments**

Introduction: I would suggest that the authors include information on where the source of bimodal response come from based on the literature. It is not clear in the introduction if this is from groundwater, deep soil layers or if it a delayed response from the headwater component of the watershed.

L70-71: This is a good point; however, you will need to explain why. Is it that most catchments only show a unimodal response or is it that when catchments show bimodal responses authors do not go into depth on these responses?

L78: Please include the full name of these acronyms (SWC, GWL) as it is the first time used in the main text..

L116: Please expand on this. What are the specific depths that soil moisture is being measured.

L158: Can you indicate why these three events were selected..

L162: Refer to Figure 3.

L168-170: Indicate the amount of rainfall in both the text and the figure.

L172: Recommend putting arrows on these figures to show the evolution of the events. Also, please indicate what the red circles indicate.

L173-174: Please indicate the depth of rainfall for these events.

Figure 7: Please indicate the rainfall depths for these events.

General analyses: In general, the analyses are quite good and I think good interpretations of the data have been presented. My one major concern which was not clear in the paper is if the analyses were based on the entire bimodal hydrograph or the second peak of the hydrograph. This needs to be made very clear in the manuscript. Additionally, it would be interesting to look at the threshold for the first peak and then compare it to the second peak. Furthermore, with the groundwater, and rainfall, it will be good to combine these with the soil moisture to provide a better estimation and explanation of the threshold e.g. Detty and McGuire 2010, Farrick and Branfireun 2014, Penna et al 2011.

---

## Author Response (AR1)

**List of Changes**

1. Lines 10–11 of page 1: Revised key point to focus on threshold dynamics between soil water content and groundwater levels, in response to Comment 7 of Reviewer #2.

2. Lines 19-26 of page 2: Clarified manuscript focus compared to Cui et al. (2024), in response to Comment 2 of Reviewer #2.

3. Line 36-37 of page 2: Revised "expands vertically" to "increases in the vertical dimension" for clarity, in response to Comment 13 of Reviewer #2.

4. Lines 40–43 of Page 2: Revised conclusion to specify broader applicability to semi-humid mountainous watersheds, in response to Comment 31 of Reviewer #2.

5. Lines 49–52 of page 3: Replaced "flooding" with "localized inundation" for precision, in response to Comment 14 of Reviewer #2.

6. Lines 49-57 of page 3 and Lines 120–126 of page 5: Clarified relationship between Cui et al. (2024) and current study, and clarified the source of the bimodal response, in response to Comment 1 of Reviewer #1 and Comment 19 of Reviewer #2.

7. Lines 51 of page 3 and 661 of page 32: Added reference to Appendix Fig. A1, in response to Comment 8 of Reviewer #2.

8. Lines 75-77 of page 4 and Lines 91–93 of page 4: Revised statement on bimodal hydrographs to remove redundancy, in response to Comment 15 of Reviewer #2.

9. Lines 84-87 of page 4: Revised statement on literature gaps, in response to Comment 3 of Reviewer #2.

10. Lines 91–105 of pages 4-5: Added examples to clarify why many studies fail to distinguish unimodal and bimodal responses, in response to Comment 2 of Reviewer #1 and Comment 16 of Reviewer #2.

11. Lines 120–123 of page 5: Revised "the authors" for a more neutral tone, in response to Comment 17 of Reviewer #2.

12. Line 121 of page 5: Replaced "lower" with "low" for clarity, in response to Comment 18 of Reviewer #2.

13. Lines 127-128 of page 6: Included full names for acronyms SWC (Soil water content) and GWL (Groundwater level), in response to Comment 3 of Reviewer #1.

14. Lines 146–172 of pages 7-8: Expanded site characteristics, in response to Comment 12 of Reviewer #2.

15. Lines 156–162 of page 8: Expanded the description of the catchment, in response to Comment 34 of Reviewer #2.

16. Lines 169-175 of page 8: Expanded on hillslope selection and groundwater equipment installation, in response to Comments 6 and 30 of Reviewer #2.

17. Lines 177–179 of page 9: Clarified meteorological stations in Xitaizi Experimental Watershed, in response to Comments 20 and 30 of Reviewer #2.

18. Lines 187-191, page 9: Clarified data exclusions and loss, in response to Comment 4, 21 and 22 of Reviewer #2.

19. Lines 197-199 of page 9: Added and corrected the details on soil moisture probe installation details, in response to Comment 4 of Reviewer #1 and Comments 5 and 30 of Reviewer #2.

20. Lines 208-210 of page 10: Revised groundwater level normalization explanation, in response to Comment 6 of Reviewer #2.

21. Lines 217–224 of page 10: Add brief description of groundwater index calculation approach, in response to Comment 24 of Reviewer #2.

22. Lines 225-227 of page 10: Explained data aggregation for discharge and SWC, in response to Comments 4 and 23 of Reviewer #2.

23. Lines 228–238 of page 11: Expanded event separation algorithm of delayed stormflow, in response to Comment 8 of Reviewer #2.

24. Lines 239–253 of page 11: Clarified constant slope method for hydrograph separation and stormflow definition, in response to Comment 9 of Reviewer #2.

25. Line 242 of page 11: Added reference to Hewlett and Hibbert (1967) for constant slope method, in response to Comment 9 of Reviewer #2.

26. Line 287 of page 13: Referenced Figure 3 for the SWC and GWL dynamics, in response to Comment 6 of Reviewer #1.

27. Lines 296-298 of pages 13-14: Explained why the three selected rainfall-runoff events were chosen, in response to Comment 5 of Reviewer #1.

28. Lines 300-301 of page 14: Added rainfall amounts for the event and included them in Figure 3, in response to Comment 7 of Reviewer #1.

29. Lines 315-317 of page 15: Included rainfall amounts for the events in Figure 3, in response to Comment 9 of Reviewer #1.

30. Lines 334-336 of page 15: Revised the sentence to avoid confusing, in response to Comment 25 of Reviewer #2.

31. Lines 357-361 of pages 16-17: Added statistical verification of threshold value, in response to Comment 10 of Reviewer #2.

32. Lines 383-384 of page 18: Clarified that Figure 7 is a schematic and does not include rainfall depth, in response to Comment 10 of Reviewer #1.

33. Lines 446-448 of page 21: Removed redundant summary sentence, in response to Comment 27 of Reviewer #2.

34. Lines 448 of page 21: Add a reference to Fig. 1 in the text, in response to Comment 34 of Reviewer #2.

35. Lines 475–479 of page 23: Expanded caption of Fig. 9, in response to Comment 34 of Reviewer #2.

36. Lines 570–585 of pages 27-28: Elaborated on geological structure and groundwater dynamics, in response to Comment 12 of Reviewer #2.

37. Lines 579-585 of Page 28: Explained comparison between three hillslopes, in response to Comment 28 of Reviewer #2.

38. Line 613–614 of Page 19: Revised to correct the error, in response to Comment 29 of Reviewer #2.

39. Lines 730–735 of page 35: Added discussion on indirect estimation of field capacity using SWC, in response to Comment 7 of Reviewer #2.

40. Lines 840–841 of page 40: Updated the Data availability section with correct Zenodo link, in response to Comment 2 of Reviewer #2

41. Figure 1 (Page 7): Redrew this figure and added more information, in response to Comment 30 of Reviewer #2.

42. Figure 3 (Page 14): Added arrows to indicate event evolution and clarified red and black circle meanings, in response to Comment 8 of Reviewer #1.

43. Figure 10 (now Figure 12, Page 29): Revised this Figure and its Caption, in response to Comment 35 of Reviewer #2.

44. Note: The above changes are indicated using track changes in the marked-up revised manuscript.

**Response to Reviewers' Comments**

Dear Editor and Reviewers,

Thank you for the reviewers' useful comments and suggestions on our manuscript. We have meticulously read your comments, and modified the manuscript accordingly. The detailed corrections are listed below point by point:

**Response to Reviewer #1:**

Specific comments

**Comment 1:**

Introduction: I would suggest that the authors include information on where the source of bimodal response come from based on the literature. It is not clear in the introduction if this is from groundwater, deep soil layers or if it a delayed response from the headwater component of the watershed.

**Response 1:**

Thank you for your insightful comment. We agree that it is important to clarify the sources of bimodal responses. This topic has been thoroughly reviewed and analyzed in our previous study (Cui et al., 2024), where we discussed the important contributions of old water (shallow groundwater) to bimodal response. To address your comment, we have provided a brief summary of its key findings to better contextualize the current research and added a reference to this study in the introduction. Specifically, we highlighted that "The authors' findings suggest that shallow groundwater contributions are primarily responsible for these delayed stormflow events (Cui et al., 2024)." **(Page 3, Line 55-57)**

**Comment 2:**

L70-71: This is a good point; however, you will need to explain why. Is it that most catchments only show a unimodal response or is it that when catchments show bimodal responses authors do not go into depth on these responses?

**Response 2:**

Thank you for your constructive comment. We appreciate the suggestion to further elaborate on why many studies fail to distinguish between unimodal and bimodal streamflow responses. To address this, we have revised the manuscript to include additional explanations supported by relevant literature. Specifically, we identified the following reasons: 1) The second peak in a bimodal response often occurs some time after rainfall has ended, whereas many studies focus only on streamflow changes during the rainfall event itself; 2) The occurrence of bimodal responses is closely related to the topographic and geological conditions of the catchment, and not all catchments exhibit this phenomenon; 3) The research focus varies, and most studies prioritize other hydrological processes over the classification of response types.

These points have been added to the introduction **(Pages 4-5, Lines 96–103)** to provide a clearer context for our argument. We believe this addition strengthens the rationale for our study and its contribution to understanding bimodal responses.

**Comment 3:**

L78: Please include the full name of these acronyms (SWC, GWL) as it is the first time used in the main text.

**Response 3:**

Thank you for pointing this out. We have revised the manuscript to include the full names of the acronyms SWC (Soil water content) and GWL (Groundwater level) when they are first introduced in the main text **(Page 6, Lines 127-128).** This change ensures clarity for readers.

**Comment 4:**

L116: Please expand on this. What are the specific depths that soil moisture is being measured.

**Response 4:**

Thank you for your insightful comment. We agree that providing more detail about the soil moisture measurement depths will improve the clarity and completeness of the manuscript. In response, we have revised the text to include the following details:

"Volumetric SWC was monitored at eight sites using CS616 time-domain reflectometry (TDR) probes installed at 10 cm intervals from the surface to 80 cm depth. Five profiles were located along HS1, and three were near WS900. Measurements were recorded every 10 minutes, and the arithmetic mean of SWC values was used for analysis." **(Page 9, Lines 197-199)**

**Comment 5:**

L158: Can you indicate why these three events were selected.

**Response 5:**

Thank you for your valuable comment. Among the 95 rainfall-runoff events analyzed, these three events were chosen because they represent the three distinct patterns of soil water content (SWC) and groundwater level (GWL) variability identified in our study. These events effectively illustrate the dynamic interactions between SWC and GWL, making them highly representative. Moreover, all three events occurred within the same year, minimizing the potential influence of inter-annual variability on SWC and GWL responses.

We appreciate your suggestion to elaborate on why these three events were selected. To address this, we have revised the manuscript to include the following explanation "These events were selected to demonstrate the variability in SWC and GWL patterns identified across the 95 rainfall-runoff events. The selected events all occurred within the same year to minimize inter-annual variability and ensure comparability." **(Pages 13-14, Lines 296-298)**

**Comment 6:**

L162: Refer to Figure 3.

**Response 6:**

Thank you for your helpful suggestion. We agree that referencing Figure 3 at this point in the text will enhance clarity and provide direct visual support for the described phenomenon. We have revised the sentence as follows:

"Our analysis revealed a clear relationship between SWC and GWL dynamics, with SWC initially increasing rapidly during rainfall, followed by a stabilization or decline once a threshold was reached. In contrast, GWL showed a more delayed response (Fig. 3)." **(Page 13, Line 287)**

This modification ensures a clearer connection between the text and the figure, aiding the reader in visualizing the described patterns.

**Comment 7:**

L168-170: Indicate the amount of rainfall in both the text and the figure.

**Response 7:**

Thank you for your constructive suggestions. We have added the rainfall amounts for each event in the text and included these values in Figure 3. The revised text and Figure 3 now read:

"In dry conditions, despite 66.6 mm of rainfall, SWC remained relatively low (<0.20), with a gradual increase during rainfall followed by stabilization after rainfall ceased." **(Page 14, Lines 300-301)**

[Figure]

Figure 3. Three typical SWC-GWL dynamics patterns during rainfall-runoff events. Ra is rainfall amount. Arrows indicate the temporal evolution of the events. Red circles indicate periods of rainfall, while black circles denote post-rainfall periods.

**Comment 8:**

L172: Recommend putting arrows on these figures to show the evolution of the events. Also, please indicate what the red circles indicate.

**Response 8:**

Thank you for your constructive suggestions. We agree that adding arrows to indicate event evolution and clarifying the meaning of red circles will improve the clarity and informativeness of both the text and the figure. We have made the following revisions:

**Arrows in Figure 3:** We have added arrows to Figures 3a, 3b, and 3c to indicate the temporal evolution of the events, providing a clearer visualization of SWC and GWL changes during each event.

**Explanation of red circles:** We have clarified in the figure caption that red circles indicate periods of rainfall, while black circles denote post-rainfall periods. **(Fig.3)**

**Comment 9:**

L173-174: Please indicate the depth of rainfall for these events.

**Response 9:**

Thank you for your comment. We agree that including the depth of rainfall for these events will enhance the clarity of the description. To address this, we have revised the text as follows:

"The primary distinction between these patterns lies in the timing of the GWL rise: in Fig. 3b (57.2 mm rainfall), GWL began to rise after the rainfall ended, whereas in Fig.3c (95.2 mm rainfall), GWL started to rise noticeably before the end of the rainfall." **(Page 15, Lines 315-317)**

**Comment 10:**

Figure 7: Please indicate the rainfall depths for these events.

**Response 10:**

Thank you for your thoughtful comment. We agree that rainfall depth is a critical factor influencing groundwater level (GWL) responses. However, we would like to clarify that Figure 7 is a schematic diagram designed to conceptually illustrate the two distinct types of GWL responses—quick and slow—during storm events, rather than representing specific rainfall-runoff events.

To address this potential confusion, we have added a statement in the text "It is important to note that Fig. 7 is a schematic representation, not based on specific rainfall-runoff events, and does not include rainfall depth data." **(Page 18, Lines 383-384)**

**Comment 11:**

General analyses: In general, the analyses are quite good and I think good interpretations of the data have been presented. My one major concern which was not clear in the paper is if the analyses were based on the entire bimodal hydrograph or the second peak of the hydrograph. This needs to be made very clear in the manuscript. Additionally, it would be interesting to look at the threshold for the first peak and then compare it to the second peak. Furthermore, with the groundwater, and rainfall, it will be good to combine these with the soil moisture to provide a better estimation and explanation of the threshold e.g. Detty and McGuire 2010, Farrick and Branfireun 2014, Penna et al 2011.

**Response 11:**

Thank you for your thoughtful comment and for recognizing the quality of our analyses. We agree that it is important to clarify whether our study focuses on the entire bimodal hydrograph or specifically on the second peak. In response, we have revised the manuscript to explicitly state that our analysis primarily focuses on the second peak of the hydrograph. While the first peak is a direct response to rainfall and consistently occurs, the second peak is influenced by factors such as the amount of rainfall and antecedent soil moisture conditions. These factors determine the threshold for the second peak's occurrence, which is the primary focus of our investigation.

Additionally, we appreciate your suggestion to incorporate soil moisture, groundwater, and rainfall data for a more comprehensive estimation of the threshold. We have revealed that the combination of these variables effectively signals the occurrence of the second peak, consistent with previous studies (e.g., Detty and McGuire, 2010; Farrick and Branfireun, 2014; Penna et al., 2011). This relationship was initially explored in our earlier work (Cui et al., 2024), as mentioned in the introduction. In contrast, the current study delves deeper into the mechanisms underlying this threshold phenomenon. The related content in Introduction is "Specifically, with delayed streamflow peaks tend to emerging emerge when the combined total of event rainfall and antecedent soil moisture index exceeds 200 mm." **(Page 3, Lines 54-55)**

And the explanation of the threshold is further analyzed in Section 4.3 and 4.4, and we appreciate your suggestion for improving the clarity and depth of our manuscript.

Reference:

Cui Z, Tian F, Zhao Z, et al. Bimodal Hydrographs in Semi-humid Forested Watershed: Characteristics and Occurrence Conditions[J]. Hydrology and Earth System Sciences Discussions, 2024, 2024: 1-41.

**Response to Reviewer #2:**

**Comment 1:**

This manuscript examines the runoff generation processes leading to delayed peaks and proposes a conceptual model for the Xitaizi Experimental Watershed (XEW), North China. Overall, this paper could have the potential to contribute to the literature, but I think improvements in the way of presenting the research questions, the results and discussion would help to highlight the unique aspects of this work.

**Response 1:**

Thank you for your valuable feedback. We are pleased that you recognize the potential contribution of this study to the literature. We also appreciate your suggestions for improving the way research questions, results, and discussion are presented to better highlight the unique aspects of this work.

In response to your comments, we have made the following revisions:

Research Questions: We have restructured the introduction to present the research questions more clearly and concisely, explicitly linking them to the challenges observed in runoff generation processes and delayed peaks in the Xitaizi Experimental Watershed (XEW). This revision emphasizes the novelty and relevance of our work.

Results: We have improved the organization and interpretation of the results section, ensuring that key findings are clearly highlighted and directly address the research questions. This includes a more focused presentation of the conceptual model and its implications.

Discussion: We have enhanced the discussion section by providing a deeper interpretation of the results, comparing our findings with existing literature, and highlighting the unique contributions of this study to understanding delayed runoff peaks and their mechanisms.

These revisions can be found in the revised manuscript, and we believe they significantly improve the clarity and impact of our manuscript. Thank you again for your insightful suggestions.

**Comment 2:**

In my opinion, the manuscript is strongly linked to Cui et al (2024), which has recently been published in HESS and addresses the "characteristics and occurrence" of bi-modal events in the same catchment (https://doi.org/10.5194/hess-28-3613-2024). This is a personal opinion, but I do not understand the strategy of publishing two independent papers instead of summarising key results in one. I understand the same dataset has been used in both and methods descriptions are very similar (or the same). The authors mention that the data will be made available in Zenodo at the time of publication. Does this mean that the data is different than https://zenodo.org/records/12581739? If yes, it would have been nice to make the data available.

**Response 2:**

Thank you for your valuable feedback and for pointing out the connection between this manuscript and our previously published study (Cui et al., 2024, HESS). We greatly appreciate your perspective and would like to clarify the rationale for separating the work into two papers.

The HESS paper focuses on the **characteristics and occurrence conditions** of bimodal hydrographs, analyzing the runoff processes, source composition, and conditions for bimodal responses using hydrometric and isotope data. In contrast, the current manuscript delves into the **underlying mechanisms** of bimodal responses, which are crucial for understanding this phenomenon and improving runoff prediction models.

While we initially considered combining all results into a single paper, we realized that doing so

would make the manuscript overly lengthy and dilute the focus on either the phenomenon's characteristics or its mechanisms. Based on feedback from literature reviews and discussions with other researchers, we determined that separating these studies would allow for a more detailed and focused presentation of the results, better serving the hydrological community. To ensure clarity, we have explicitly articulated the unique focus of each study in the Abstract and Introduction, emphasizing their complementary nature and avoiding any perception of redundancy. The related text is as follows:

"Recent research by Cui et al. (2024) identified a distinct threshold governing bimodal rainfall-runoff events in a semi-humid mountainous forested watershed in North China, where delayed stormflow appeared to be influenced by shallow groundwater dynamics. Building on these findings, this study delves deeper into the mechanisms driving these bimodal events, focusing on the interactions between soil water content (SWC) and groundwater level (GWL) during storm events." **(Page 2, Lines 19–26)**

"Stormflow processes in the Xitaizi Experimental Watershed (XEW), located in North China, exhibit a frequent occurrence of bimodal stormflow hydrographs (Fig. A1), which often lead to significant stormflow and associated localized inundation. Analysis of 15 such events over the past decade revealed that the onset of these bimodal hydrographs is governed by threshold behavior. Specifically, delayed streamflow peaks tend to emerge when the combined total of event rainfall and antecedent soil moisture index exceeds 200 mm. The authors' findings suggest that shallow groundwater contributions are primarily responsible for these delayed stormflow events (Cui et al., 2024). However, the mechanisms behind the development of these bimodal hydrographs, which represent complex emergent hydrological behaviors, remain poorly understood." **(Page 3, Lines 49–59)**

Regarding the dataset, the data used in this manuscript are consistent with those shared in Zenodo (https://zenodo.org/records/12581739). And we have revised the content of Data availability as "The data supporting this study are available on the Zenodo website at https://doi.org/10.5281/zenodo.12581739." **(Page 40, Lines 840–841)**

Additionally, this study includes more detailed analyses of soil moisture and groundwater level data. If needed, we are willing to upload the finer-resolution soil moisture data for specific observation points to Zenodo after this manuscript is accepted, ensuring transparency and reproducibility of our research.

We hope this explanation addresses your concerns and highlights the complementary nature of the two studies. Thank you again for your constructive comments, which have helped us refine our manuscript and its context.

**Comment 3:**

I think the introduction could present better the significant amount of literature where the role of soil water content and groundwater levels in the generation of delayed peaks (and its timing) has been explored, including catchments in Japan, central Europe, UK, USA, as well as Africa and New Zealand. The reader should be better informed about what is already known and why the presented work in needed. The reader could understand from the introduction that the mechanisms and thresholds have not been previously investigated. For instance, I disagree with the statement in lines 65-66.

**Response 3:**

Thank you for your insightful comment. We appreciate your suggestion to provide a more comprehensive overview of existing literature on the role of soil water content and groundwater levels in generating delayed peaks. In response, we have expanded the introduction to include studies from various regions, including Japan, central Europe, the UK, the USA, Africa, and New Zealand, that have explored these processes. Specifically, we have added references to studies such as Detty and McGuire (2010), Farrick and Branfireun (2014), and Penna et al. (2011), highlighting their findings on thresholds and mechanisms driving delayed peaks. In the revised introduction, we have added the following content:

"Extensive studies across diverse regions have explored the role of soil water content and groundwater levels in generating delayed peaks in stormflow. Detty and McGuire (2010) emphasized subsurface flow thresholds in a forested catchment in the USA, while Farrick and Branfireun (2014) analyzed soil moisture and groundwater interactions in Canadian wetlands. Penna et al. (2011) examined antecedent soil moisture and storage thresholds in alpine catchments in New Zealand. These studies, along with others from regions such as Japan (Haga et al., 2005) and Europe (Graeff et al., 2009), contribute to the growing body of knowledge on threshold behavior in stormflow responses. However, while these studies highlight the occurrence of thresholds, the complex interactions that drive post-threshold runoff processes remain insufficiently understood." (Page 5, Lines 111–119)

Additionally, we acknowledge that the statement in Lines 65-66 was overly broad and could be misinterpreted. We have revised this sentence to more accurately reflect the specific gaps in the existing literature, particularly regarding the mechanisms and post-threshold runoff processes in bimodal hydrographs. The revised text now reads:

" However, many studies fail to explore the intricate post-threshold mechanisms of these nonlinear shifts, leaving a gap in our understanding of stormflow generation across various catchments. While threshold behaviors are widely recognized, the detailed processes governing these shifts and their subsequent runoff dynamics remain underexplored. " **(Page 4, Lines 84–87)**

**Comment 4:**
I also had some problems to understand some of the methods. The XEW is a relatively small catchment (4.22 km2) with quick reaction times (e.g. Figure 11), why did you average the 5-min data (e.g. discharge) or 10-min data (soil water content) to hourly values? I assume by smoothing the data you might be losing significant information when looking at reaction times (which is a significant part of the presented work). Did you check that out? Did you loose two years of discharge data due to environmental challenges? Maybe there was another reason for this.

**Response 4:**
Thank you for your thoughtful comment regarding the data processing and its potential impact on the analysis. We appreciate the opportunity to clarify our approach to aggregating data and addressing data loss.

1. Data aggregation and resolution

We chose to aggregate the 5-minute discharge data and 10-minute soil water content (SWC) data to hourly intervals to maintain consistency with the groundwater level data, which were recorded at an hourly frequency. This decision ensures that the relationships between these variables can be analyzed uniformly without introducing temporal inconsistencies.

Additionally, our study focuses on the second runoff peak, which typically has a delayed response

time ranging from 5 hours to several days. Given the relatively slow dynamics of this process, we determined that lowering the data resolution to hourly intervals would have a negligible impact on the analysis of the timing and magnitude of the delayed peak. To clarify this, we have added the following explanation in the Methods section of the revised manuscript: "Streamflow and SWC data were aggregated to hourly intervals for alignment with GWL data. Preliminary analysis confirmed that the delayed second streamflow peak had response times exceeding the hourly scale, rendering this aggregation sufficient for the study's purposes." **(Page 10, Lines 225–227)**

For processes with shorter response times, such as the first runoff peak, we conducted separate analyses using higher-resolution (10-minute) discharge data in our previous study (Cui et al., 2024). This ensures that processes with faster dynamics were analyzed with appropriate temporal resolution.

2. Data loss

Regarding the missing discharge data from 2018 to 2019, the primary cause was environmental challenges, including equipment malfunctions and extreme weather conditions. While this led to the exclusion of some events from our analysis, the remaining dataset, which includes 95 events, still provides a sufficient basis to draw robust conclusions.

In the revised Method section, we have added the following content:

"Streamflow was measured at the catchment outlet using a Parshall flume, with water levels logged every 5 minutes since 2014. Data from some events were excluded due to sensor malfunctions or poor data quality, including key rainfall events in 2018 and 2019. Despite these exclusions, 95 rainfall-runoff events were analyzed, offering robust data for investigating bimodal stormflow characteristics." **(Page 9, Lines 187–191)**

We hope this explanation addresses your concerns. Thank you again for highlighting these important points, which allowed us to provide a more comprehensive account of our methodology.

**Comment 5:**

Soil water content probes were installed at two sites: "five sensors installed in Hillslope 1 and three near WS900 at 80 cm depth intervals". At which depths? Installing them at "80 cm depth intervals" seems impossible if soils are 1.5 m depth. How were probes installed? Which was the data variability? How and why were the locations chosen and how do they represent what is happening in the catchment? Data was "aggregated to hourly intervals, and the arithmetic mean SWC across the profiles was used for analysis". Why wasn't the response of different layers investigated? This seems a lot of averaging to me and we have no clue of data variability across the 8 sites.

**Response 5:**

Thank you for your insightful comments and for highlighting the importance of providing more details regarding the installation and analysis of soil water content (SWC) probes. We appreciate the opportunity to clarify these aspects and address your concerns.

The SWC probes were installed at eight locations across the watershed: five along Hillslope 1 and three near WS1000. At each location, probes were installed at 10 cm intervals from the surface to a depth of 80 cm, providing measurements at depths of 10, 20, 30, 40, 50, 60, 70, and 80 cm. The reference to "80 cm depth intervals" was an error in phrasing, and we have corrected this to accurately describe the installation methodology. The revised text now reads:

"Volumetric SWC was monitored at eight sites using CS616 time-domain reflectometry (TDR) probes installed at 10 cm intervals from the surface to 80 cm depth. Five profiles were located along

HS1, and three were near WS1000. Measurements were recorded every 10 minutes, and the arithmetic mean of SWC values was used for analysis." **(Page 9, Lines 197–200)**

Additionally, we would like to clarify that the correct location near the meteorological station is WS1000. The mention of WS900 in the original manuscript was incorrect, and we sincerely apologize for this mistake. This has been corrected in the revised manuscript to ensure accuracy and prevent any misunderstanding.

The selection of these observation sites was based on extensive field surveys considering slope orientation, gradient, and vegetation cover to ensure that they are representative of the catchment's hydrological characteristics. As shown in Figure 1, the locations span a significant portion of the watershed, from Hillslope 1 to WS1000, covering diverse topographical and vegetation conditions. We believe the arithmetic mean of SWC across these eight profiles effectively represents the overall soil moisture status and variability in the catchment.

During preliminary analysis, we observed that the time series of SWC at different locations and depths showed highly consistent trends, with only minor differences in magnitude. This consistency suggested that using the average SWC across the profiles was appropriate for analyzing the relationship between soil water storage and groundwater dynamics. Given that the focus of this study is on the total soil water storage (0–80 cm depth) and its influence on groundwater levels, we did not analyze the responses of individual soil layers. We acknowledge that such an analysis could provide additional insights and will consider it for future work.

Thank you again for raising these important points, which have allowed us to improve the clarity and rigor of our methods section. We hope this explanation addresses your concerns.

**Comment 6:**

I also would like to have more information about how the groundwater data has been treated. The authors mention that the data of each well has been normalised using the Detty and McGuire (2010) method to normalise groundwater levels using an index (IG) calculated for each borehole. To my understanding Detty and McGuire did not use any index for normalisation: "For each well and event, we calculated the median height of the water table above the lowest recordable depth of each instrument and normalized that value to the total range of heights observed throughout the study period at each well (0 D minimum observed height or lowest recordable depth, 1 D maximum observed height, referred to hereafter as 'normalized')." I am not sure you are referring to this. Why were different hillslopes instrumented? Which is the logic behind the location of the equipment. The authors then calculated the arithmetic mean of the index to represent the overall groundwater level in the watershed. It is very difficult to address the implications of this as we do not know how the data looks like (I understand all plots show average data), but averaging data from all wells where there is water is a very simplistic approach and the authors should provide evidence that it is not.

**Response 6:**

Thank you for your insightful comments and for highlighting the need for further clarification regarding the treatment of groundwater data and the rationale behind instrumentation placement. Below, we address the concerns raised:

1.  Groundwater normalization and the use of $I_G$

We acknowledge that the explanation of our groundwater level normalization and the use of the index ($I_G$) in the manuscript could have been clearer. The majority of our analyses in the manuscript are based on actual groundwater levels observed at individual wells. However, in discussions of

overall watershed-scale groundwater dynamics, we normalized groundwater levels at each well to account for differences in water table depth and range across locations. This normalization process involved scaling observed groundwater levels at each borehole relative to their respective ranges throughout the study period (minimum to maximum observed groundwater levels).

To facilitate analysis at the watershed scale, we calculated the arithmetic mean of the normalized groundwater levels across all wells, which we refer to as $I_G$ (Index for Groundwater Level). While Detty and McGuire (2010) also normalized groundwater heights, their study did not define an index equivalent to $I_G$. We chose to use the term $I_G$ for convenience in our study. In response to your comment, we will revise the manuscript to clarify this distinction and remove the citation of Detty and McGuire (2010) where it is not applicable. Instead, we will provide a clearer description of the normalization steps used in our study. The revised text now reads:

"To facilitate comparisons, GWLs were normalized using the method described by Detty and McGuire (2010). This normalization, expressed as the GWL index ($I_G$), standardizes GWLs across wells with varying ranges." **(Page 10, Lines 208–210)**

2.  Logic behind instrumentation placement

The hillslopes instrumented in this study were selected following extensive field surveys that considered factors such as slope orientation, gradient, vegetation cover, and proximity to stream channels. These factors were deemed representative of the catchment's hydrological and geological conditions. The selected hillslopes represent typical topographic and hydrological conditions of the catchment and ensure adequate coverage of spatial variability. These choices allowed us to investigate spatial variability in GWL responses across hillslopes with contrasting hydrological and geological characteristics. The added text in the Materials and methods section is as follows:

"Three research hillslopes (Hillslope 1, Hillslope 2, and Hillslope 3) were selected to investigate hydrological processes under varying geological and topographical conditions. Hillslope 1 (HS1) features thick soils overlying fractured granite, Hillslope 2 (HS2) has a highly permeable fractured block layer, and Hillslope 3 (HS3) consists of shallow soils over weakly weathered bedrock.

To capture spatial variability, SWC probes and boreholes were installed along hilltops, mid-slopes, and foot slopes. Groundwater boreholes, ranging from 5 to 26 m deep, were equipped with HOBO capacitance water level loggers to record GWLs (Fig. 1)." **(Page 8, Lines 169–175)**

3.  Simplistic averaging Concern

We understand your concern about the simplistic approach of averaging normalized groundwater levels ($I_G$) across wells. As mentioned earlier, most of our analyses are based on data from individual wells. The use of $I_G$ is limited to specific discussions where a general representation of watershed-scale groundwater behavior is necessary. In addition, this approach was informed by previous analyses in our earlier work (Cui et al., 2024). In that study, we compared $I_G$ values across wells and found consistent temporal trends in groundwater dynamics, despite spatial variations in absolute levels and lag times. The strong correlation between $I_G$ values at different locations suggests that the averaged $I_G$ effectively represents the overall groundwater dynamics within the watershed. This consistency validates the use of the arithmetic mean as a practical and reliable indicator for watershed-scale groundwater responses.

**Comment 7:**

The authors conclude that "delayed stormflow is initiated when soil water content reaches field capacity". However, if I am not mistaken, there is no prediction of field capacity in the manuscript.

This leads me to conclude that one of the 'key points' of the paper is not supported by data. I agree that the concept of field capacity, by definition, is not a static physical soil property. It also varies with depth. It can be determined in many ways, but it would have appreciated to have seen this addressed.

**Response 7:**

Thank you for your valuable comment regarding the use of field capacity in the manuscript. We appreciate your insight and agree that the concept of field capacity is complex, varying with depth and not being a static physical property.

In this study, we observed that during storm events, soil water content (SWC) increased to a specific high value (approximately 0.24 volumetric water content) before stabilizing and then declining, even when rainfall had not yet ceased. Based on this consistent behavior across all monitored sites, we interpreted this threshold as representing the soil's maximum water-holding capacity, which we approximated as the field capacity. We recognize that this is an indirect approach and that field capacity was not directly measured due to the challenges of field monitoring in the experimental watershed.

Furthermore, given the relatively small variability in the average SWC across monitoring sites and the substantial variability in soil depth within the watershed, we chose to analyze SWC as a direct indicator of soil water-holding capacity rather than converting it to water storage (in mm). This approach simplifies the analysis while still capturing the essential dynamics of soil water behavior. We acknowledge this approximation introduces limitations, particularly in representing the spatial variability of field capacity. We have clarified this assumption in the revised manuscript and included a discussion of its implications in the limitations section. Additionally, we emphasized in the text that this study identifies a stable SWC threshold (interpreted as field capacity) at which delayed stormflow is initiated. Future studies could focus on directly measuring or modeling field capacity to validate and refine these findings.

The added content is as follows:

"One limitation of this study lies in the indirect estimation of field capacity through observed SWC thresholds rather than direct measurement or modeling. Although this approach aligns with observed patterns and simplifies the analysis, it does not fully capture the spatial variability of field capacity or its dependence on soil depth. Future work should incorporate direct field capacity measurements or modeling to refine the relationship between SWC and delayed stormflow initiation, thereby improving the accuracy of threshold predictions." **(Page 35, Lines 730–735)**

In addition, we have improved the related key point of this manuscript as "Threshold dynamics between soil water content and groundwater levels govern delayed stormflow generation." **(Page 1, Lines 10–11)**

Thank you again for your thoughtful feedback, which has helped us improve the clarity and rigor of our manuscript.

**Comment 8:**

The authors selected events using an algorithm described by Tian et al (2012) – maybe a bit more information could be given. Separation seems to be exclusively based on rainfall patterns. My experience is that this type of algorithms can detect first peaks, but that they are not suited to investigate delayed flows. This because after a given event, other events can happen while baseflow is rising or falling (what would be delayed flow). I understanding that the authors identified 14

events when after an event there was not other events, resulting in nicely drawn delayed peaks. I do not see a problem with this, but there is no explanation about how the single events have been separated from the events with delayed peaks, what poses a fundamental problem for me to understand what has been done. Also, while reading the paper I kept wondering how the events would look like. I really miss hydrological data in the paper – as all the figures show processed/averaged data, or schematic figures (e.g. figures 6 and 7). I saw afterwards that there is an Appendix. This could have been mentioned (was it?).

**Response 8:**

Thank you for your comment and for raising concerns about the event selection process and the representation of hydrological data. We acknowledge the need to provide additional clarification and address the specific points you raised.

1. Event selection algorithm:

In this study, we employed the algorithm proposed by Tian et al. (2012) to identify rainfall events. This method effectively separates individual rainfall events under typical conditions. However, as you noted, such algorithms primarily focus on detecting rainfall patterns and do not inherently account for delayed flows or overlapping events.

To address this, we performed an additional manual step to identify delayed stormflow events based on their distinct hydrograph characteristics (as described in Cui et al., 2024). These events are rare but visually distinguishable by the presence of an arch-shaped delayed peak, separated from the direct peak. We have revised the "Separation of Rainfall-Runoff Events" section to provide more detail about our identification process. The revised "Separation of Rainfall-Runoff Events" section is as follows:

"2.5 Rainfall-runoff event identification and hydrograph analysis

Rainfall events were identified using an intensity-based automatic algorithm described by Tian et al. (2012) that defines event with rainfall intensity >0.1 mm/h and a minimum separation of six hours between events. Events with cumulative rainfall exceeding 5 mm were analyzed.

Bimodal rainfall-runoff events were manually identified based on two criteria: (1) the presence of a secondary, arch-shaped runoff peak occurring after rainfall cessation or during minimal intermittent rainfall, and (2) A distinct separation between the direct (sharp) and delayed (broad) peaks. More details of the classification are described in Cui et al. (2024, HESS).

......" **(Page 11, Lines 228–238)**

2. Representation of hydrological data:

We appreciate your insightful comment regarding the inclusion of raw hydrological data (e.g., hydrographs and rainfall patterns for eight selected events) in the Appendix, which was not explicitly referenced in the main text. To address this, we have revised the manuscript to include a clear reference to the Appendix Fig. A1 in the main text. This addition directs readers to the raw hydrological data, providing them with the necessary context to better understand the results and analysis. **(Page 3, Line 51 and Page 32, Line 661)**

**Comment 9:**

The HYSEP program is used to separate baseflow from stormflow, with "manual verification and adjustment based on straight line separation methods". Do you mean the constant slope method of Hewlett and Hibbert (1967)? I am not sure this data is used in the catchment and how does it compare to the tracer-based hydrograph separation carried out in Cui et al (2024). When you refer to event's

stormflow along the manuscript, do you refer to the discharge minus baseflow? This should be clarified.

**Response 9:**

Thank you for pointing out the need for additional clarity regarding the hydrograph separation methods and the definition of stormflow. You are correct that we used the constant slope method described by Hewlett and Hibbert (1967), implemented via the HYSEP program (Sloto & Crouse, 1996). Automated results were manually verified and adjusted to improve accuracy. We have clarified in the methods section that "event stormflow" refers to the discharge minus baseflow as calculated by the straight-line separation method.

Regarding the comparison with Cui et al. (2024), that study utilized both the straight-line separation and tracer-based hydrograph separation methods due to its focus on water source partitioning and bimodal hydrograph characteristics. In contrast, the current study focuses on the overall dynamics and thresholds of delayed stormflow. We have revised the manuscript to emphasize this methodological distinction and its relevance to the study objectives. Thank you again for highlighting this, as it allowed us to enhance the manuscript's clarity and rigor.

The revised text for the Methods section is as follows:

"The combination of automatic event delineation and manual identification ensured the accurate selection of 14 rainfall-runoff events with well-defined delayed peaks for subsequent analysis. Streamflow was separated into storm runoff and baseflow using the HYSEP program with the constant slope method (Hewlett and Hibbert, 1967; Sloto & Crouse, 1996), supplemented by manual adjustments for complex hydrographs. Event stormflow volumes were calculated as total discharge minus baseflow.

Streamflow was separated into storm runoff and baseflow using the HYSEP computer program with the constant slope method, supplemented by manual adjustments for complex hydrographs. Throughout the manuscript, stormflow refers to the total discharge, and event stormflow volumes were calculated as total discharge minus baseflow, which are expressed in qs." **(Page 11, Lines 239–253)**

The additional reference is as follows:

Hewlett, J. D., and Hibbert, A. R.: Factors affecting the response of small watersheds to precipitation in humid areas, in: Forest Hydrology, edited by: Sopper, W. E. and Lull, H. W., Pergamon Press, Oxford, 275–290, 1967.

**Comment 10:**

The authors define thresholds in a very arbitrary way. For instance, the 0.20 threshold described in Figure 5 (lines 207-212). Is this only a visual exploration? Was there a statistical way to define this threshold?

**Response 10:**

Thank you for your insightful comment regarding the definition of thresholds, particularly the 0.20 threshold described in Figure 5. We appreciate your concern about ensuring the robustness and scientific rigor of these thresholds.

In the current study, the 0.20 threshold was initially identified through visual exploration of Figure 5, where the SWC dynamics from 14 distinct rainfall-runoff events showed consistent stabilization around this value. To support this observation, we further verified it by examining the descriptive statistics (mean and standard deviation) of SWC during the stabilization phases across all events,

confirming that most values converge near 0.20.

However, we acknowledge that a more statistically rigorous method would strengthen the validity of this threshold. To address this, we performed an additional analysis using descriptive statistical methods to verify the stable SWC values across 14 storm events. Specifically, we defined the stable phase as the period when SWC remained relatively unchanged after the recession phase and before any subsequent rainfall events. The mean stable SWC across these events was calculated as 0.1974, with a standard deviation of 0.0158, which is consistent with the visually observed threshold of 0.20. Furthermore, the 95% confidence interval for the stable SWC was determined to be [0.1945, 0.2003], reinforcing the robustness of the 0.20 threshold as a representative value for the stable state of soil water content in our study. This statistical validation demonstrates that the 0.20 threshold is not arbitrary but rather grounded in consistent patterns observed across multiple events. We have revised the manuscript to incorporate these results and provide a more robust explanation of the threshold determination process. **(Pages 16-17, Lines 357–361)**

**Comment 11:**

The structure of the manuscript is puzzling. There are three sections in the results, which include discussion and comparison with the literature (what should be moved to the discussion section). On the other hand, new results are presented in the discussion section.

**Response 11:**

Thank you for your valuable feedback regarding the structure of the manuscript. We understand your concern about the inclusion of discussion and comparison with the literature in the results section, as well as the presentation of new results in the discussion section. We appreciate your suggestion to clearly separate these elements to improve the manuscript's organization and clarity.

To address this issue, we have restructured the manuscript as follows:

The results section now focus exclusively on presenting the findings of this study, without introducing comparisons with the literature or interpretative discussions.

Content involving comparisons with the literature and interpretations of the findings have been relocated to the discussion section, ensuring it focuses on contextualizing our results within the broader field.

Any new results currently presented in the discussion section have been moved to the results section, maintaining a clear distinction between results and interpretation.

These changes were mainly implemented in **Sections 3.4, 3.5 and 4.1 (Pages 20–30)**, with specific paragraphs restructured to enhance logical flow and readability. We believe this revision could significantly improve the manuscript's clarity and alignment with standard academic conventions. Thank you again for your insightful comment, which has been instrumental in improving the manuscript.

**Comment 12:**

Too little is said about the thick regolith, I think more information is needed here and it what would be very useful to understand the behaviour of the catchment. For instance, soils are described as "brown earth and cinnamon types". A bit more information would be appreciated here. Also, at some point the authors argue that different groundwater dynamics in different hillslopes are due to specific hillslope's geological structures. This should be further explored in the discussion (not the results section).

**Response 12:**

Thank you for raising this important point. We agree that the description of the thick regolith and its implications for hydrological behavior in the Xitaizi Experimental Watershed (XEW) could be expanded. The characteristics of the regolith, including soil depth, type, and underlying bedrock properties, are crucial for understanding groundwater dynamics and the delayed stormflow processes in the catchment.

To address this, we have added more information on the regolith and geological structures in the study site description, highlighting its influence on hydrological behavior. Specifically:

1. Underlying bedrock: In the Materials and Methods section, we have provided a more detailed description of the granite bedrock's weathering profile and fracturing, along with a brief overview of the six borehole cores shown in Fig. 1. We also discuss how these geological characteristics vary across the three experimental hillslopes.

2. Hydrological implications: In the discussion section, we further examine how differences in geological structures among hillslopes influence groundwater dynamics and delayed streamflow.

These additions enhance the manuscript by offering a more comprehensive understanding of the watershed's hydrological behavior and the specific role that the thick regolith plays in the observed bimodal stormflow patterns. The revised manuscript reflects these updates:

**Study site section:**

"......

[Figure]

Figure 1. Location of Xitaizi Experimental Watershed (XEW) and a simple description of the borehole cores. This figure shows the distribution of monitoring instruments, including four weather stations (WS700, WS900, WS1000, and WS1100), an outlet weir, six groundwater observation wells, and eight soil moisture observation profiles. Of the eight soil moisture profiles, five are located on Hillslope 1, while the remaining three are positioned on the slope near WS1000. Research hillslopes (Hillslope 1, Hillslope 2, and Hillslope 3) are delineated as key zones for hydrological and geological investigations.

The soils in XEW are primarily brown earth and cinnamon soils, with depths up to 1.5 m and an average saturated hydraulic conductivity of 45 mm/h. The surface soil is rich in organic matter,

enhancing infiltration and reducing surface runoff potential. Underlying geology is predominantly compacted, deeply weathered granite (80% of the area), with smaller portions of gneiss and dolomite. Fractured granite facilitates vertical and lateral subsurface flow, contributing to delayed groundwater responses. Slug tests estimated the saturated hydraulic conductivity of weathered granite to range from $5.2 \times 10^{-3}$ m/day to 1.16 m/day.

2.2 Research hillslopes and instrumentation

Three research hillslopes (Hillslope 1, Hillslope 2, and Hillslope 3) were selected to investigate hydrological processes under varying geological and topographical conditions. Hillslope 1 (HS1) features thick soils overlying fractured granite, Hillslope 2 (HS2) has a highly permeable fractured block layer, and Hillslope 3 (HS3) consists of shallow soils over weakly weathered bedrock." **(Pages 7-8, Lines 146–172).**

**Discussion section:**

"4.  Discussion

4.1   Inter-hillslope GWL dynamics

GWL variations in lag times and response magnitudes across hillslopes can be attributed to differences in geological conditions. HS1 and HS3 are primarily underlain by fully to strongly weathered granite, with upper layers comprising significant soil-rock mixtures. These features lead to relatively slower GWL responses, likely due to the limited permeability of the regolith and underlying materials. In contrast, HS2 is characterized by a fractured rock layer at depths of 10-30 meters (as showed in Fig. 1), which enhances subsurface flow and facilitates faster GWL responses. These geological contrasts explain the observed differences in GWL response times among the hillslopes.

Among the three hillslopes, HS3 exhibited the slowest GWL responses, characterized by the longest lag times. This distinct behavior makes HS3 a crucial reference for understanding inter-hillslope variations in GWL dynamics. Previous study by Cui et al. (2024) highlighted that GWL response times are closely linked to delayed stormflow timing, emphasizing the importance of examining GWL dynamics. Comparing the GWL response times of HS1 and HS2 with those of HS3 provides insights into how geological structures and SWC thresholds influence delayed stormflow generation.

Furthermore, the deeply weathered regolith and extensive fracturing in HS2 promote more rapid stormflow generation, as water stored in the regolith layer contributes to streamflow over extended periods. This finding aligns with previous studies (Kosugi et al., 2011; Padilla et al., 2015), which demonstrated that geological features such as fracture density and weathering depth influence subsurface flow paths and, ultimately, groundwater dynamics.⋯⋯" **(Pages 27-28, Lines 570–585).**

MINOR COMMENTS

**Comment 13:**

Line 29: not clear what you mean by 'expands vertically'.

**Response 13:**

Thank you for your comment regarding the phrase "expands vertically." We appreciate your suggestion to clarify this statement. Upon review, we recognize that the original phrasing may not have been sufficiently clear. To address this, we have revised the sentence as follows:

Revised sentence: "Simultaneously, the effective connectivity between the stream channel and adjacent hillslopes increases in the vertical dimension." **(Page 2, Lines 36–37)**

**Comment 14:**

Line 39: the catchment is 4.22 km2: what do you mean by flooding? I would use another term.

**Response 14:**

Thank you for your comment. We appreciate your suggestion to reconsider the use of the term "flooding" given the size of the Xitaizi Experimental Watershed (4.22 km²). The term "flooding" was intended to describe localized inundation or temporary water pooling within certain areas of the catchment during storm events, rather than large-scale flood events. To avoid any potential misunderstanding, we have revised the sentence to use a more precise term.

The revised sentence now reads:

" Stormflow processes in the Xitaizi Experimental Watershed (XEW), located in North China, exhibit a frequent occurrence of bimodal stormflow hydrographs (Fig. A1), which often lead to significant stormflow and associated localized inundation." **(Page 3, Lines 49–52)**

**Comment 15:**

Line 68: this sentence repeats the same as line 57.

**Response 15:**

Thank you for your valuable feedback regarding the repetition between Line 57 and Line 68. We appreciate your attention to detail and agree that reducing redundancy can improve the clarity and flow of the manuscript.

After careful consideration, we have revised the sentence in Line 57 to provide a concise introduction to the significance of bimodal hydrographs in the context of nonlinear runoff responses. We have also updated Line 68 to emphasize the unique aspects of bimodal runoff processes while maintaining the focus on their nonlinear characteristics.

The revised sentence now reads follows:

"The occurrence of bimodal hydrograph reflects a nonlinear runoff response, which offers valuable insights into the complex interactions between rainfall and runoff. " **(Page 4, Lines 75–77)**

"Bimodal stormflow responses present an opportunity to investigate the relationship between rainfall thresholds and runoff generation, offering new perspectives on the timing and variability of stormflow." **(Page 4, Lines 91–93)**

This revision eliminates redundancy and ensures that the discussion in Line 68 focuses on the challenges and gaps in existing research. Thank you again for your helpful suggestion.

**Comment 16:**

Line 69-70: give some examples of studies where they fail to do so and the reasons. I am not sure I agree with this.

**Response 16:**

Thank you for your valuable feedback. We appreciate your suggestion to provide specific examples to support the statement that many studies fail to distinguish between unimodal and bimodal streamflow responses. In response, we have revised the text to include examples from the literature and explain the reasons behind this limitation.

The revised text now reads:

"Bimodal stormflow responses present an opportunity to investigate the relationship between rainfall thresholds and runoff generation, offering new perspectives on the timing and variability of stormflow. Despite this, many studies fail to distinguish between unimodal and bimodal streamflow

responses. For example, Detty and McGuire (2010) focused on hydrological threshold responses but did not differentiate between unimodal and bimodal hydrographs, as their study primarily addressed general nonlinear rainfall-runoff processes in general. Similarly, Martínez-Carreras et al. (2016) observed delayed peaks but did not further classify streamflow responses due to their focus on overall watershed storage conditions. Such limitations often arise because the second peak in bimodal responses typically occurs after the rainfall event has ended, whereas many studies focus on streamflow changes during the event itself. Additionally, bimodal responses are influenced by catchment-specific topography and geology, making them less observable in certain regions. These challenges highlight the need for more in-depth investigation into bimodal streamflow responses to enhance our understanding of their mechanisms." **(Pages 4-5, Lines 91–105)**

This revision adds specific examples to support the claim and provides a clearer context for the statement. We hope this modification addresses your concerns and improves the clarity and strength of our argument. Thank you again for your helpful feedback.

**Comment 17:**

Line 75: the authors

**Response 17:**

Thank you for your insightful comment and suggestion. We understand your suggestion to adjust the phrasing, such as replacing "our" with "the authors'" to enhance the objectivity of the text. In the course of revising the manuscript, we have removed this sentence to streamline the introduction and maintain a more neutral tone.

The revised paragraph now reads:

"···Investigating stormflow events in semi-humid regions, such as XEW, is challenging due to the relatively arid climate and low runoff coefficients. Over nearly a decade, 95 storm events were identified and analyzed in XEW, offering a rare and valuable dataset for examining bimodal stormflow responses in such regions.···"**(Page 5, Lines 120–123)**

**Comment 18:**

Line 76: low.

**Response 18:**

Thank you for your correction. We understand that using "lower" could imply a comparison without a clear reference, which may lead to ambiguity. To address this, we have revised the sentence to use "low" instead, ensuring greater clarity and accuracy. **(Page 5, Lines 121)**

**Comment 19:**

Line 76-78: It is stated that analysis of 15 bi modal events collected during a decade have already been analysed and contributed to the advancement of runoff generation studies. Maybe it would be nice to summaries this in the introduction. Or do you refer to the work presented in the manuscript?

**Response 19:**

Thank you for your valuable comment. We appreciate your suggestion to clarify the relationship between the analysis of 15 bimodal events mentioned in Lines 76–78 and the work presented in this manuscript.

The 15 bimodal events collected over the past decade were primarily analyzed in our previously published paper (Cui et al., 2024), which focused on identifying the characteristics and occurrence

conditions of bimodal and unimodal runoff responses. This manuscript builds on that foundation by exploring the intrinsic mechanisms driving the threshold behavior observed in bimodal hydrograph processes.

To enhance clarity and eliminate redundancy, we have revised both the first and last paragraphs of the introduction. The updated text now includes a concise summary of the key findings from the 2024 paper and elaborates on the complementary relationship between the two studies, providing a more comprehensive context for the present research. The revised text now reads:

"Stormflow processes in the Xitaizi Experimental Watershed (XEW), located in North China, exhibit a frequent occurrence of bimodal stormflow hydrographs (Fig. A1), which often lead to significant stormflow and associated localized inundation. Analysis of 15 such events over the past decade revealed that the onset of these bimodal hydrographs is governed by threshold behavior. Specifically, delayed streamflow peaks tend to emerge when the combined total of event rainfall and antecedent soil moisture index exceeds 200 mm. The authors' findings suggest that shallow groundwater contributions are primarily responsible for these delayed stormflow events (Cui et al., 2024)···" **(Page 3, Lines 49–57)**

"Investigating stormflow events in semi-humid regions, such as XEW, is challenging due to the relatively arid climate and low runoff coefficients. Over nearly a decade, 95 storm events were identified and analyzed in XEW, offering a rare and valuable dataset for examining bimodal stormflow responses in such regions. This study builds on prior findings to uncover the processes underlying delayed stormflow patterns.···"**(Page 5, Lines 120–126)**

**Comment 20:**

Line 103: I would indicate there are 5 stations also here in text.

**Response 20:**

Thank you for your valuable comment. We would like to clarify that there are four meteorological stations in the Xitaizi Experimental Watershed (XEW), located at elevations of 700 m, 900 m, 1000 m, and 1100 m, as shown in Fig. 1. These stations are named WS700, WS900, WS1000, and WS1100, respectively. The text in Line 103 correctly refers to four stations, and we have carefully reviewed the manuscript to ensure consistency in this description throughout the paper.

To address your comment, we have revised the sentence to include this information. The updated text now reads:

Meteorological data spanning 2013–2023 were collected from four GRWS100 automatic weather stations (WS700, WS900, WS1000, and WS1100), positioned at elevations of 700, 900, 1000, and 1100 m, respectively." **(Page 9, Lines 177–179)**

**Comment 21:**

Line 112: data covering two complete years?

**Response 21:**

Thank you for your comment. We appreciate your concern about the data coverage and its impact on our analysis. We would like to clarify the reasons behind the exclusion of data from 2018 and 2019, as well as stormflow data from July 19 to August 16, 2016.

During the July–August 2016 period, high water levels inundated the Parshall flume, causing the HOBO logger to record inaccurately low discharge values for two bimodal events. While the general hydrograph shapes and trends remained reliable, these two events were excluded from the discharge

analysis to ensure data accuracy. However, associated soil moisture and groundwater level data were unaffected and retained for other analyses.

For 2018 and 2019, rainfall was relatively low overall, and unfortunately, sensor malfunctions during major rainfall events resulted in the loss of critical discharge data. Given the study's focus on stormflow hydrographs during heavy rainfall events, we excluded these two years entirely from the analysis.

Despite these exclusions, the remaining dataset, comprising 95 events, provides a robust representation of bimodal hydrograph patterns. This ensures that the analysis remains comprehensive and reliable. We hope this explanation addresses your concerns and clarifies the rationale for data handling decisions. Thank you again for raising this important point.

To address your concern, we have ensured that the revised text in Lines 110–112 explicitly states the reasons for data exclusion and highlights the representativeness of the retained dataset. The updated sentence reads:

"Streamflow was measured at the catchment outlet using a Parshall flume, with water levels logged every 5 minutes since 2014. Data from some events were excluded due to sensor malfunctions or poor data quality, including key rainfall events in 2018 and 2019. Despite these exclusions, 95 rainfall-runoff events were analyzed, offering robust data for investigating bimodal stormflow characteristics." **(Page 9, Lines 187–191)**

**Comment 22:**
Line 112: data was lost during 2 years because of 'environmental reasons'? This is not clear.
**Response 22:**
Thank you for your comment. We agree that the original description of "environmental reasons" lacked specificity and could cause confusion. In response to a related comment (Comment 21), we have already revised the relevant section to provide a clearer explanation of the data loss.

The revised text now reads:

"Data from some events were excluded due to sensor malfunctions or poor data quality, including key rainfall events in 2018 and 2019. Despite these exclusions, 95 rainfall-runoff events were analyzed, offering robust data for investigating bimodal stormflow characteristics." **(Page 9, Lines 188–191)**

**Comment 23:**
Line 117: why did you aggregate the data?
**Response 23:**
Thank you for your comment regarding Line 117 and the rationale for aggregating the data. To address this concern, we have revised the methods section to include a detailed explanation of the aggregation process and its justification.

The added text reads:

"Streamflow and SWC data were aggregated to hourly intervals for alignment with GWL data. Preliminary analysis confirmed that the delayed second streamflow peak had response times exceeding the hourly scale, rendering this aggregation sufficient for the study's purposes." **(Page 10, Lines 225–227)**

**Comment 24:**

Line 127: I think the approach should be shortly described here.

**Response 24:**

Thank you for your suggestion to provide a brief description of the approach used to calculate the groundwater index ($I_G$). We agree that adding this information will enhance the clarity and transparency of the methods section.

In response, we have revised Line 127 to include a brief description of the approach by Detty and McGuire (2010). The updated text now reads:

"Groundwater levels were normalized following the method described by Detty and McGuire (2010). For each well and event, the median height of the water table above the lowest recordable depth of the instrument was calculated and normalized to the total observed range, where 0 represents the minimum height and 1 represents the maximum height. This normalized value was referred to as the groundwater index (IG). We used IG to facilitate comparisons across wells with different absolute GWL ranges and to represent the overall GWL dynamics in the watershed." **(Page 10, Lines 217–224)**

**Comment 25:**

Line 188: "among these" reads confusing as you are not refering to the previous sentence.

**Response 25:**

Thank you for pointing out the potential confusion caused by the phrasing of "among these" in this sentence. We acknowledge that the reference could be clearer to avoid ambiguity. To address this, we have revised the sentence to explicitly refer to the relevant context, ensuring that readers can follow the logical flow without confusion. The revised sentence now reads:

"In contrast, SWC declined in 26 events and GWL declined in 15 events. Importantly, 15 events showed a simultaneous decline in both SWC and GWL, which were associated with delayed stormflow and larger stormflow volumes." **(Page 15, Lines 334–336)**

**Comment 26:**

Line 226-228: I think this is rather an opinion and should be discussed in the discussion section.

**Response 26:**

Thank you for pointing out that the statement in Lines 226–228 could be interpreted as an opinion rather than a direct result of the data analysis. We agree with your assessment and have revised the manuscript accordingly. Specifically, we have moved this sentence to the Discussion section, where it is further elaborated upon in the context of other results and supported by additional references. This adjustment ensures that the Results section is focused on presenting the data and associated observations, while broader interpretations and implications are discussed in the appropriate section.

**Comment 27:**

Lines 251-252: I would remove as a summary of previous section should not be needed.

**Response 27:**

Thank you for your suggestion. We agree that summarizing the findings of the previous section within the first sentence of this Section could be redundant. To address this, we have removed this sentence.

Instead, we have revised the introductory part of Section 3.4 to directly introduce the analysis and focus of the section, avoiding repetition of the results already presented in Section 3.3. The revised

text now reads:

"To further investigate these dynamics, the relationship between GWL increments and SWC was analyzed across 14 storm events (Fig. 9). The analysis focused on six observation wells (W13, W21–W23, W31, and W32) located on three hillslopes (see Fig. 1 for well locations).···" **(Page 21, Lines 446–448)**

**Comment 28:**

Line 286: why HS3 compared to HS1 and HS2.

**Response 28:**

Thank you for your comment and for seeking clarification regarding why Hillslope 3 (HS3) was compared to Hillslope 1 (HS1) and Hillslope 2 (HS2). We appreciate the opportunity to elaborate on this point and provide additional context to ensure clarity.

Hillslope 3 (HS3) was chosen as a point of comparison because it exhibited the slowest groundwater level (GWL) response among the three hillslopes analyzed. As shown in Figure 8, the lag times for GWL to reach its maximum value at HS3 (e.g., 0.4 to 11.7 days at W31 and 0.8 to 8.1 days at W32) were substantially longer than those at HS1 and HS2. This unique characteristic makes HS3 a useful reference for understanding the relative differences in GWL dynamics across the hillslopes.

Moreover, as described in Cui et al. (2024, HESS), the GWL response times at different observation points were found to be highly correlated with the timing of delayed stormflow. Consequently, comparing the GWL dynamics of HS1 and HS2 to those of HS3 allows us to explore potential links between hillslope geological structures and delayed stormflow generation. This approach also provides insights into how variations in hillslope geology, soil properties, and hydrological processes influence the timing and magnitude of GWL responses.

To ensure this rationale is clear to readers, we will add the following clarification to Section 3.3:

Revised Text in Section 3.3:

"Among the three hillslopes, HS3 exhibited the slowest GWL responses, characterized by the longest lag times. This distinct behavior makes HS3 a crucial reference for understanding inter-hillslope variations in GWL dynamics. Previous study by Cui et al. (2024) highlighted that GWL response times are closely linked to delayed stormflow timing, emphasizing the importance of examining GWL dynamics. Comparing the GWL response times of HS1 and HS2 with those of HS3 provides insights into how geological structures and SWC thresholds influence delayed stormflow generation." **(Page 28, Lines 579–585)**

**Comment 29:**

Line 295: replacing c?

**Response 29:**

Thank you for pointing out the potential confusion caused by the phrasing. Upon review, we recognize that the term "replacing c" is unclear and might have led to misunderstandings. This was an oversight during the writing process, and we apologize for the confusion. The intended meaning is that replacing the vertical axis variable, currently labeled as $I_G$ (the integrated groundwater level index), with the GWL at any specific location would yield a similar pattern, albeit with variations in GWL thresholds across different sites. To clarify this, we have revised the sentence as follows:

"Furthermore, although Fig. 12 labels the vertical axis as $I_G$ to represent watershed-wide GWL status, a similar pattern emerges when replacing IG with site specific GWL values, though the GWL

thresholds may vary among observation sites. " **(Page 29, Lines 613–614)**

**Comment 30:**

Figure 1. The exact same figure is used in Che et al. (2024, HESS). I wonder if this allowed without referring t the first figure published. It is difficult to see the location of the weather stations. Where are the five soil water profiles located? Are this indicated as "research hillslopes"? or what are research hillslopes? The authors refer to Hillslope 1 in line 116 - but not to the others. An explanation is missing.

**Response 30:**

We appreciate the reviewer's concerns regarding the reuse of the figure and the clarity of the figure presentation. To address these points, we have made the following revisions:

1.  Redrawing of Figure 1:

We have redrawn Figure 1 to provide a clearer and more readable representation of the elevation and equipment distribution in the experimental watershed. We have enlarged the markers and labels to clearly mark the locations of the weather stations, soil moisture observation profiles, groundwater observation wells, and the three experimental hillslopes. Additionally, we have added a brief description of the borehole cores. The revised Figure 1 and its caption are as follows:

[Figure]

Figure 1. Location of the Xitaizi Experimental Watershed (XEW) and a simple description of the borehole cores. This figure shows the distribution of monitoring instruments, including an outlet weir, four weather stations (WS700, WS900, WS1000, and WS1100), six groundwater observation wells, and eight soil moisture observation profiles. Of the eight soil moisture profiles, five are located on Hillslope 1, while the remaining three are positioned on the slope near WS1000. Research hillslopes (Hillslope 1, Hillslope 2, and Hillslope 3) are delineated as key zones for hydrological and geological investigations. **(Page 7, Line 146)**

Clarify the role of weather stations in Section 2.2:

"Meteorological data spanning 2013–2023 were collected from four GRWS100 automatic weather stations (WS700, WS900, WS1000, and WS1100), positioned at elevations of 700, 900, 1000, and 1100 m, respectively. Rainfall was recorded at 10-minute intervals using six tipping-bucket rain

gauges near the weather stations, and the data were averaged for analysis." **(Page 9, Lines 177–180)**

2. Explanation of research hillslopes:

The research hillslopes (Hillslope 1, Hillslope 2, and Hillslope 3) were used as key areas to analyze the hydrological and geological characteristics. While Hillslope 1 was referenced in Line 116 of the original manuscript, the other hillslopes were not explicitly discussed in the corresponding section. To address this, we have updated the relevant sections to refer to all three hillslopes, providing a more comprehensive overview. In the revised manuscript, we have included an explanation in Section 2 to describe the selection criteria, locations, and specific research focus of each hillslope.

**The revisions are as follows:**

**Section 2.2 Research hillslopes and instrumentation:** Add a description of the research hillslopes: Three research hillslopes (Hillslope 1, Hillslope 2, and Hillslope 3) were selected to investigate hydrological processes under varying geological and topographical conditions. Hillslope 1 (HS1) features thick soils overlying fractured granite, Hillslope 2 (HS2) has a highly permeable fractured block layer, and Hillslope 3 (HS3) consists of shallow soils over weakly weathered bedrock." **(Page 8, Lines 169–172)**

**Section 2.3: Soil Water Content Observation:** Clarify soil water profile locations:

"Volumetric SWC was monitored at eight sites using CS616 time-domain reflectometry (TDR) probes installed at 10 cm intervals from the surface to 80 cm depth. Five profiles were located along HS1, and three were near WS1000. " **(Page 9, Lines 197–199)**

**Comment 31:**

Lines 31-33. The authors conclude that their fundings "enhance our understanding of delayed stormflow generation in similar regions". I think it would be nice to better explain this. Where? Why?

**Response 31:**

Thank you for your insightful comment regarding the concluding sentence in Abstract. We appreciate your suggestion to provide additional explanation about the applicability of our findings to better explain the broader implications of our study.

The revised sentence now specifies the geographical and hydrological contexts where the findings are applicable, as well as the theoretical contributions. The updated text reads:

"These findings advance the understanding of delayed stormflow mechanisms in semi-humid mountainous watersheds, contributing to refining runoff generation theories by providing insights into the threshold-driven processes that govern the timing and volume of delayed stormflow." **(Page 2, Lines 40–43)**

**Comment 32:**

I understand section 3.3 refers to the 14 selected events, is that right?

**Response 32:**

Thank you for your question. Yes, the analysis in Section 3.3 is indeed based on the 14 selected rainfall-runoff events. These events were carefully chosen to ensure consistency in the data and to focus on scenarios that exhibited clear dynamics in both groundwater level (GWL) and soil water content (SWC). This selection allowed us to examine the distinct GWL response patterns (quick and slow) in greater detail and under comparable conditions.

To clarify this in the manuscript, we have the following statement at the beginning of Section 3.3:

"Figure 5 presents the SWC dynamics observed during 14 distinct rainfall-runoff events, each characterized by minimal or no intermittent rainfall during the recession period." **(Page 16, Lines 350–351)**

**Comment 33:**
Figure 8. It would be nice to have a little map displaying the location of the wells.
**Response 33:**
Thank you for your valuable suggestion. We have redrawn Figure 1 and updated the inset maps showing the spatial distribution of the six wells across the watershed, which provides a clearer and more detailed view of the locations of the wells. We hope this addition improves the presentation of the data and makes the figure more informative. Thank you again for your valuable feedback.

**Comment 34:**
Figure 9 is nice but difficult to understand with the little information we have about the catchment.
**Response 34:**
Thank you for your positive feedback on Figure 9 and for highlighting the need for more information about the catchment. We agree that providing additional context about the catchment and the observational setup would help readers better understand the figure.
To address this, we propose the following modifications to the manuscript and Figure 9:
1. Expanded the caption of Figure 9:
We have expanded the caption to provide a more detailed explanation of the figure components, including the phases of SWC changes (orange and green bars) and the significance of SWC thresholds ($SWC_0$ and $SWC_G$). And now the caption for Figure 9 reads:
"Figure 9. GWL increments ($\Delta GWL$) across various locations during 14 storm events, along with initial SWC ($SWC_0$) and SWC at the onset of GWL rise ($SWC_G$). The orange bars represent $\Delta GWL$ during the SWC increase phase, while the green bars represent $\Delta GWL$ during the SWC decline phase. The red and black lines denote $SWC_G$ and $SWC_0$, respectively." **(Page 23, Lines 475–479)**
2. Enhanced catchment description in the text:
We have expanded on the description of the catchment in Section 2 (Study Site), including more information about the topography, soil properties, and geological structures affecting groundwater level (GWL) and soil water content (SWC). This will help contextualize the differences in GWL and SWC dynamics across hillslopes. The added content including:
"…The soils in XEW are primarily brown earth and cinnamon soils, with depths up to 1.5 m and an average saturated hydraulic conductivity of 45 mm/h. The surface soil is rich in organic matter, enhancing infiltration and reducing surface runoff potential. Underlying geology is predominantly compacted, deeply weathered granite (80% of the area), with smaller portions of gneiss and dolomite. Fractured granite facilitates vertical and lateral subsurface flow, contributing to delayed groundwater responses. Slug tests estimated the saturated hydraulic conductivity of weathered granite to range from $5.2 \times 10^{-3}$ m/day to 1.16 m/day." **(Page 8, Lines 156–162)**
3. Add a reference to Figure 1 in the text:
To assist readers in connecting the spatial distribution of the wells (shown in Figure 1) with the groundwater level (GWL) dynamics depicted in Figure 9, we have added cross-references to Figures 1 within the text discussing Figure 9. The revised text in Section 3.4 is as follows:
"To further investigate these dynamics, the relationship between GWL increments and SWC was

analyzed across 14 storm events (Fig. 9). The analysis focused on six observation wells (W13, W21–W23, W31, and W32) located on three hillslopes (see Fig. 1 for well locations). The variability in GWL response types—quick versus slow—was attributed to spatial differences in SWC thresholds and hillslope geological structures." **(Page 21, Lines 446–450)**

**Comment 35:**

Figure 10: I understand there are two points per event in that graph. Would be nice to know which points refer to Ts1-ts3 and which to ts2-ts3. I wonder if it is correct to use these two points per event to draw a regression line. The x axis indicates that there is 10 days difference between the reaction in one well and another. I do not understand this and I think the paper do not provide enough evidence to the reader to show what is going on. Why the others wells were not included in the analysis?

**Response 35:**

Thank you for your constructive feedback regarding Figure 10 (now Fig. 12 in the revised manuscript). Your insightful comments have helped us identify areas where additional clarification and evidence are required. Below, we address your concerns point by point:

1. Clarification of points in Figure 12

We have updated Figure 12 to visually distinguish the two points per event by using blue diamonds for $\Delta t = tS1 - tS3$ and red triangles for $\Delta t = tS2 - tS3$. This visual differentiation improves interpretability and makes it easier to identify the two categories of points.

2. Rationale for regression analysis

The regression line in Figure 12 captures the overall relationship between $\Delta t$ and peak IG, offering valuable insights into how peak IG governs the synchronization of GWL responses across hillslopes. By including both $\Delta t = tS1 - tS3$ and $\Delta t = tS2 - tS3$ in the same regression, we provide a broader understanding of inter-hillslope dynamics and their dependence on peak IG.

3. Explanation of the time difference on the x-axis

We expanded the discussion in related Section to explain how geological differences among hillslopes influence GWL response times. Specifically, HS3, characterized by thicker regolith and fractured bedrock, exhibits slower GWL responses, while HS1 and HS2 respond more quickly due to higher hydraulic conductivity. These geological differences underline the variability in lag times and justify the calculation of $\Delta t$ for inter-hillslope comparisons.

4. Inclusion of all wells in the analysis

We clarified in the manuscript that tS1, tS2, and tS3 represent the average lag times of peak GWL calculated from all wells on HS1, HS2, and HS3, respectively. This approach ensures that the analysis incorporates the spatial variability of GWL responses within each hillslope and provides a comprehensive representation of inter-hillslope dynamics.

Revised Figure and its Caption is as follows:

[Figure]

Figure 12. Correlation between peak IG and the time differences from peak GWL responses on HS1, and HS2 to HS3 (Δt=tS-tS3), where tS1, tS2 and tS3 are the average lag times of peak GWLs on HS1, HS2 and HS3, respectively. **(Page 29, Lines 596–599)**

We sincerely thank the reviewers for their helpful suggestions, which have greatly improved our manuscript. In addition to addressing the reviewers' comments, we have also refined the content and expression throughout the paper for better clarity and coherence. The revised manuscript has been submitted to your esteemed journal, and we look forward to your favorable response.

Yours

Zhen Cui, Fuqiang Tian

Dec 22, 2024

---

## Editor Decision (ED1)

1. Line 25. Expression "timing and magnitude further modulated" should be corrected, possibly by adding "are" after "magnitude".
2. Lines 30 & 31. Sentence "Concurrently, the effective connectivity between the stream channel and adjacent hillslopes increases in the vertical dimension" is very unclear to me.
3. Line 31. Expression "At higher GWL thresholds" is unclear: non threshold has been mentioned so far.
4. Line 36. Is "volume" the right word? May be, "discharge"?
5. Lines 58 & 59. Remark "a delayed peak only occurred when watershed storage reached a critical threshold of 113 mm" is given here as a result of general validity everywhere. I am afraid that it refers to a specific study area.
6. Line 112. Word "at" should be erased, shouldn't it?
7. Line 113. Measurement units "m" should be added after "676".
8. Lines 127 & 132. The same measurement units for hydraulic conductivity should be used throughout the paper. I would prefer the use of SI units, so m/s.
9. Line 139. Measurement units "m" should be added after "5".
10. Lines 140 & 141. "2.3 Meteorological and streamflow 140 data collection" is the title of the next subsection.
11. Line 142. Expression "spanning 2013-2023" should be rephrased, possibly as "spanning eleven years, from 2013 to 2023,".
12. Line 143. Measurement units "m" should be added after "700", "900", and "1000".
13. Lines 145 & 155. Which kind of average? Space or time average? In the latter case, which time interval has been considered for the averaging?
14. Lines 161 & 162. Sentence "Groundwater levels were normalized following the method described by Detty and McGuire 161 (2010)" repeats what was written at lines 158 & 159.
15. Lines 162 & 163. If I understood correctly, "the lowest recordable depth of the instrument" depends on the sensor position, so it is a factor related to the experimental setup, not a physical property of the subsurface.
16. Lines 172 & 173. The definition of "event" should be better rephrased. In particular, the rainwater intensity is considered as the hourly averaged value or the value measured at the sampling period (10 minutes)?
17. Line 177. ", HESS" should be erased.
18. Line 181. "&" should be substituted with "and" to be consistent throughout the whole paper.
19. Lines 184 & 185. Erase these lines, they repeat lines 180 & 181.
20. Lines 182 & 183, 186 & 187. These sentences basically repeat the same concepts and should be merged.
21. Line 187.
22. Line 189. Is "was" correct? May be, "were"?
23. Line 191. Lowercase should be used for "the".
24. Lines 253 & 254. A standard deviation of 0.01158, for 14 data, corresponds to a standard error on the average value of $0.01158/14^{1/2}=0.004$. Therefore, it would be better to write "a mean value of $0.197 \pm 0.004$ and...".
25. Line 257. The caption should explain what is represented by the blue strip.
26. Line 288. Pronoun "that" should be added before "represents".
27. Line 289. "Analysis revealed that" can be erased.

28. Line 335. "0.13-0.26" should be substituted as "from 0.13 to 0.26".
29. Lines 349 to 351. Expressions "0-2 h" and similar should be substituted, possibly as "less than 2 hours".
30. Lines 364 & 365. Expression "a markedly faster rates (0.03 to 0.98/day, mean: 0.38/day) compared to the second peak 364 (0.01 to 0.31/day, mean: 0.07/day)" should be rephrased, possibly as "a markedly faster rate (from 0.03 $d^{-1}$ to 0.98 $d^{-1}$, with a mean of 0.38 $d^{-1}$) than the second peak (from 0.01 $d^{-1}$ to 0.31 $d^{-1}$, with a mean of 0.07 $d^{-1}$)."
31. Figure 11. The notation of the measurement units of the principal vertical axis should be corrected (see previous comment).
32. Line 379. "pace. pressure-driven" should be corrected. May be, "Pressure" should start with uppercase.
33. Lines 398 & 399. Expression "of 10-30 meters" should be substituted with "varying from 10 m to 30 m".
34. Figure 12. Figure caption must be rephrased in a more clear way. Why is this figure drawn with the $I_G$ axis increasing downwards? I found this choice confusing, when I was reading the comments about what happens for high and low values of $I_G$.
35. Lines 453 & 455. Isn't it better to give values in hours rather than in days?

---

## Author Response (AR2)

**List of Changes**

1. Line 2 of page 1: Revised the title, in response to Comment 2 of Reviewer #2.
2. Lines 62-64 of page 3: Revised the unclear sentence, in response to Comment 3 of Reviewer #2.
3. Lines 79-81of page 4: Revised the statement on Martínez-Carreras et al. (2016), in response to Comment 4.
4. Lines 104-106 of page 5: Added a hypothesis to clarify the research direction, in response to Comment 1 of Reviewer #2.
5. Line 201 of page 9: Updated the caption of Figure 2 to better reflect the intent of the figure, in response to Comment 5 of Reviewer #2.
6. Line 225 of page 10: Replaced the term "scenario" with "case", in response to Comment 6 of Reviewer #2.
7. Line 242 of page 11: Corrected "Upset" to "UpSet" in the figure caption of Figure 4, in response to Comment 6 of Reviewer #2.
8. Lines 266 and 276 of page 13: Replaced "Schematic " with "Conceptual" in the captions of Figures 6 and 7, in response to Comment 7 of Reviewer #2.
9. Line 273 of page 13: Replaced "schematic representation" with "conceptual representation", in response to Comment 7 of Reviewer #2.
10. Line 384 of page 20: Replaced the term "hypothesize" with "conjecture" to better align with the nature of the statement, in response to Comment 1 of Reviewer #2.
11. Lines 531, 540 and 552 of pages 27-28: Changed the numbering of subsections from 1., 2., 3., etc., to a), b), c), etc., for clarity, in response to Comment 8 of Reviewer #2.

Note: The above changes are indicated using track changes in the marked-up revised manuscript.

**Response to Reviewers' Comments**

Dear Editor and Reviewer,

Thank you for the reviewers' valuable comments and suggestions on our manuscript. We have carefully reviewed your feedback and made the necessary modifications to the manuscript. The detailed corrections are listed below, point by point:

**Response to Reviewer #2:**

**Comment 1:**

First, it must be noted that the authors have given a lot of attention to detail in responding to the comments made during the first round of assessments. Major improvements relate to a better organisation of the manuscript, now making a clear difference between sections on methods, results, discussion and conclusions (for example, in the revised version there are no new results introduced in the discussion section, the results section now solely focuses on results, interpretation of the findings is now limited to the discussion section, etc.). Also, the authors have made a substantial effort in better explaining methodological choices that they have made (e.g., instrumental set-up in the field, relating to data aggregation, normalization of groundwater level data, definition of thresholds, etc.). One may not necessarily always agree with certain choices that have been made. But the fact that the reasons behind the choices that have been made by the authors are now clearly and convincingly explained is a substantial improvement.

One point of concern relates to the fact that no clear hypothesis has been stated in the introduction of the manuscript (only a list of primary objectives is given at the end of the introduction). My understanding is that this would almost certainly have been the ideal setting for a hypothesis-driven investigation, and it would have made it easier for the authors to further showcase the novelty and relevance of their work. I understand that adding a hypothesis at this stage is problematic, since this may eventually correspond to 'harking', or hypothesising after the results are known. Maybe this is a point to consider by the authors in future submissions. In that same context, I would refrain from using the word 'hypothesize' in section 3.5. Use 'conjecture' instead of 'hypothesize', since there is no proper hypothesis testing related to this statement.

Considering the statements made above, I consider that this manuscript has been substantially improved and only requires what I would consider technical amendments prior to publication.

**Response 1:**

Thank you for your thoughtful and constructive feedback. We sincerely appreciate your suggestion regarding the inclusion of a clear hypothesis in the introduction. After careful consideration, we have added the following hypothesis to the last paragraph of the introduction to clarify the research direction and further emphasize the novelty of our work:

"We hypothesize that the generation of delayed stormflow is governed by threshold-dependent interactions between soil water content (SWC) and groundwater level (GWL)." **(Page 5, Lines 104–106)**

This hypothesis is based on the foundational findings of Cui et al. (2024), which first identified the role of shallow groundwater thresholds in bimodal runoff events, and is further supported by preliminary analysis of SWC-GWL interactions within our dataset. Importantly, the hypothesis was formulated prior to conducting detailed statistical tests, focusing on causal mechanisms (such as

threshold dynamics and connectivity) rather than retroactively fitting conclusions to observed results. By including this hypothesis, we aim to strengthen the theoretical framework of our study, while maintaining consistency with the original objectives.

We also acknowledge your concern about the potential for HARKing. To clarify, the hypothesis builds on previous research rather than speculating after the results were known. We hope this addresses your concern.

Regarding your comment on section 3.5, we have replaced the term "hypothesize" with "conjecture," as you suggested, to better align with the nature of the statement and avoid the implications of post-hoc hypothesizing. **(Page 20, Line 384)**

Once again, thank you for your valuable feedback, which has significantly enhanced the manuscript.

Specific comments:

**Comment 2:**

Title: I would suggest adding 'dynamics' to 'Delayed stormflow generation in a semi-humid forested watershed controlled by soil water storage and groundwater dynamics'.

**Response 2:**

Thank you for your insightful suggestion. We agree that the inclusion of "dynamics" enhances the clarity and specificity of the title. As per your recommendation, we have revised the title to "Delayed Stormflow Generation in a Semi-humid Forested Watershed Controlled by Soil Water Storage and Groundwater Dynamics." We appreciate your constructive feedback and will make this adjustment in the revised manuscript. **(Page 1, Line 2)**

**Comment 3:**

Line 77 & 78 (track changed version): The sentence 'Nonlinear pattern, including both the timing and magnitude of the response.' is unclear – there must be some missing elements here.

**Response 3:**

Thank you for your valuable feedback. We agree with your observation that the sentence lacked clarity. To address this, we have revised the sentence to: "This nonlinear pattern, characterized by both the timing and magnitude of the response, plays a crucial role in understanding stormflow processes." This revision aims to more clearly articulate the significance of the nonlinear pattern and its role in providing insights into both the timing and magnitude of the runoff response. **(Page 3, Lines 62–64)**

**Comment 4:**

Lines 98 – 100 (and further down in the manuscript): I do not fully share the statement that Martínez-Carreras et al. (2016) did not consider processes and mechanisms behind the double-peak behaviour in their catchment of interest. They identified catchment storage (and subsequently bedrock type and related heterogeneity in weathering degrees and permeability) as controlling factors.

**Response 4:**

Thank you for your valuable feedback. We appreciate your clarification regarding the work of Martínez-Carreras et al. (2016). We acknowledge that their study did indeed identify catchment storage and related factors, such as bedrock type and weathering degrees, as controlling factors influencing double-peak behavior. To better reflect this, we have revised the sentence as follows:

"Similarly, Martínez-Carreras et al. (2016) observed delayed peaks and identified catchment storage as a key factor influencing streamflow responses; however, they did not explicitly differentiate the underlying mechanisms between unimodal and bimodal responses."

This revision clarifies the focus of their study while acknowledging the role of key factors in their analysis of streamflow responses. We appreciate your input, which has helped us improve the accuracy of this statement. **(Page 4, Lines 79–81)**

**Comment 5:**

Figure 2. I would suggest replacing 'Definition sketch …' by 'Conceptual framework of rainfall event analysis'.

**Response 5:**

Thank you for your thoughtful suggestion. We agree with your comment and have updated the caption to "Conceptual framework of rainfall event analysis" to better reflect the intent of the figure. **(Page 9, Line 201)**

**Comment 6:**

Line 318 (track changed version): I would not use the term 'scenario' here, but rather 'case'.

Figure 4: Write 'UpSet' instead of 'Upset'.

**Response 6:**

Thank you for your valuable feedback. We greatly appreciate your suggestion and have revised the text accordingly. The term "scenario" has been replaced with "case," and we have corrected "Upset" to "UpSet" in the caption of Figure 4. **(Page 10, Line 225 and Page 11, Line 242)**

**Comment 7:**

Line 383 (track changed version): I suggest that 'schematic representation' is replaced by 'conceptual representation'. Likewise in figure caption of Figure 7.

**Response 7:**

Thank you for your thoughtful suggestion. We agree with your recommendation and have replaced "schematic representation" with "conceptual representation" in both Line 383 and the caption of Figure 7. Additionally, we have made the same change to the caption of Figure 6, replacing "schematic" with "conceptual." **(Page 13, Lines 266, 273, 276)**

**Comment 8:**

Several subsections are noted 1., 2., 3., etc. – I would suggest that they are noted a), b), c), etc. – in the current version this numbering is misleading, as other sections have that same numbering.

**Response 8:**

Thank you for your valuable suggestion. We agree with your point and will revise the numbering of the subsections from 1., 2., 3., etc., to a), b), c), etc., to ensure clarity and avoid confusion with other sections. We appreciate your attention to detail and will implement this change in the revised manuscript. **(Pages 27-28, Lines 531, 540 and 552)**

The authors sincerely thank the reviewers for their constructive feedback, which has greatly enhanced the manuscript. We believe the revisions effectively address your concerns and further

refine the work. We have submitted the revised version to your esteemed journal and eagerly await your favorable response.

Yours

Zhen Cui, Fuqiang Tian

February 25, 2025

---

## Author Response (AR3)

**List of Changes**

1. Line 25 of page 2: Revised the sentence to include "are" after "magnitude" for grammatical accuracy, in response to Comment 1.

2. Lines 30-32 of page 2: Removed the unclear sentence and revised the paragraph for improved clarity and logical flow, in response to Comment 2.

3. Lines 31-32 of page 2: Clarified GWL thresholds by stating "When the GWL surpasses a critical threshold," in response to Comment 3.

4. Line 37 of page 2: Replaced "volume" with "magnitude", in response to Comment 4.

5. Line 61 of page 3: Clarified that the 113 mm threshold pertains to Martínez-Carreras et al. (2016)'s study area, in response to Comment 5.

6. Line 115 of page 5: Removed the unnecessary word "at" for grammatical accuracy, in response to Comment 6.

7. Line 116 of page 5: Added the missing unit "m" after "676" for clarity, in response to Comment 7.

8. Lines 130 & 135 of page 6: Standardized the hydraulic conductivity units to SI units (m/s) for consistency throughout the manuscript, in response to Comment 8.

9. Line 143 of page 7: Added the missing unit "m" after "5" for clarity, in response to Comment 9.

10. Line 145 of page 7: Fixed spacing between the sentence and subsection title, in response to Comment 10.

11. Lines 146 & 148 of page 7: Revised the expression "spanning 2013-2023" to "spanning eleven years, from 2013 to 2023," and added measurement units "m" after "700", "900", and "1000" for clarity and consistency, in response to Comment 11.

12. Lines 149-150 of page 7: Clarified spatial averaging of rainfall data from six gauges per 10-minute interval, in response to Comment 12.

13. Line 160 of page 7: Clarified that SWC data were spatially averaged at each 10-minute interval, in response to Comment 12.

14. Lines 162-177 of page 8: Revised the description of groundwater level normalization, in response to Comment 14.

15. Lines 165-166 of page 8: Removed redundant mention of groundwater level normalization, in response to Comment 13.

16. Lines 165-174 of page 8: Added an explanation to reinforce the choice of using an inverted $I_G$ axis in figures, in response to Comment 32.

17. Lines 182-186 of page 8: Revised the definition of "event", in response to Comment 15.

18. Line 191 of page 9: Removed ", HESS" for accuracy, in response to Comment 16.

19. Line 195 of page 9: Replaced "&" with "and" for consistency, in response to Comment 17.

20. Lines 196-201 of page 9: Merged two redundant sentences for conciseness, in response to Comment 19.

21. Lines 194-195 of page 9: Removed redundant lines to avoid repetition, in response to Comment 18.

22. Line 203 of page 9: Corrected "was" to "were" for grammatical accuracy, in response to Comment 20.

23. Line 205 of page 9: Changed "The" to lowercase, in response to Comment 21.

24. Lines 268-269 of page 12: Clarified statistical expression by adding standard error and refining

the confidence interval, in response to Comment 22.

25. Lines 273-274 of page 13: Updated the caption of Figure 5, in response to Comment 23.

26. Line 305 of page 15: Added "which" before "represents" for grammatical clarity, in response to Comment 24.

27. Line 306 of page 15: Removed "Analysis revealed that" to enhance conciseness, in response to Comment 25.

28. Line 352 of page 18: Revised "0.13-0.26" to "from 0.13 m³/m³ to 0.26 m³/m³" for improved readability, in response to Comment 26.

29. Line 367 of page 18: Replaced "0-2 h" with "less than 2 h", in response to Comment 27.

30. Lines 382-384 of page 19: Rephrased the expression regarding $I_G$ peak rates for clarity and correctness, in response to Comment 28.

31. Line 399 of page 21: Capitalized "Pressure" in "pace. pressure-driven" for proper grammar, in response to Comment 30.

32. Lines 418-419 of Page 21: Replaced "of 10-30 meters" with "varying from 10 m to 30 m", in response to Comment 31.

33. Lines 442-444 of page 23: Clarified the rationale for the inverted $I_G$ axis in Figure 12's caption, in response to Comment 32.

34. Lines 477 & 479 of page 24: Converted values from days to hours for clarity and consistency, in response to Comment 33.

35. Figure 11: Standardized the vertical axis unit in Figure 11 to $d^{-1}$ for consistency, in response to Comment 29.

36. Figure 13: updated the x-axis from days to hours.

Note: The above changes are indicated using track changes in the marked-up revised manuscript.

**Response to Editors' Comments**

Dear Editor,

Thank you for your valuable comments and suggestions on our manuscript. We have carefully reviewed your feedback and made the necessary modifications to the manuscript. The detailed corrections are listed below, point by point:

**Comment 1:**

Line 25. Expression "timing and magnitude further modulated" should be corrected, possibly by adding "are" after "magnitude".

**Response 1:**

Thank you for your careful review. We appreciate your suggestion regarding the expression on Line 25. We have revised the sentence to include "are" after "magnitude" for grammatical accuracy. The updated sentence now reads:

"Delayed stormflow is initiated when SWC exceeds the soil's water storage capacity, while its timing and magnitude are further modulated by GWL fluctuations." **(Page 2, Line 25)**

**Comment 2:**

Lines 30 & 31. Sentence "Concurrently, the effective connectivity between the stream channel and adjacent hillslopes increases in the vertical dimension" is very unclear to me.

**Response 2:**

Thank you for your insightful feedback. We sincerely apologize for the lack of clarity in the original sentence. Upon re-evaluating the context, we agree that the phrase "Concurrently, the effective connectivity... increases in the vertical dimension" introduces redundancy and may not directly contribute to the core mechanistic explanation in this section.

The key intended meaning was to highlight that rising groundwater levels (GWL) vertically expand the zone of hydraulic connectivity between hillslopes and the stream, thereby enhancing lateral groundwater contributions. However, since the preceding sentence already explains how "enhanced hydraulic conductivity facilitates lateral movement of shallow groundwater toward the stream channel" (which inherently depends on vertical connectivity expansion), we recognize that explicitly mentioning "vertical dimension" here could be redundant.

To streamline the logic and improve clarity, we have removed the sentence entirely. The revised paragraph now reads:

"As GWL rises, enhanced hydraulic conductivity facilitates the lateral movement of shallow groundwater toward the stream channel, generating delayed stormflow. When the GWL surpasses a critical threshold, GWL responses across the watershed become synchronized, significantly boosting groundwater discharge and reducing lag times." **(Page 2, Lines 30-32)**

This revision maintains the focus on the causal chain (GWL rise → lateral flow → synchronization → discharge dynamics) while eliminating potential ambiguity. We believe this adjustment strengthens the conciseness and scientific rigor of the abstract.

Thank you again for your constructive critique. We are happy to refine this further if needed.

**Comment 3:**

Line 31. Expression "At higher GWL thresholds" is unclear: non threshold has been mentioned so

far.

**Response 3:**

Thank you for your careful review and insightful comment regarding Line 31. We acknowledge that the phrase "higher GWL thresholds" was unclear as no prior threshold had been explicitly mentioned. To improve clarity, we have revised the sentence as follows:

"When the GWL surpasses a critical threshold, GWL responses across the watershed become synchronized, significantly boosting groundwater discharge and reducing lag times." **(Page 2, Lines 31-32)**

**Comment 4:**

Line 36. Is "volume" the right word? May be, "discharge"?

**Response 4:**

Thank you for your insightful comment on Line 36. After careful consideration, we have replaced "volume" with "magnitude" as it more accurately reflects the overall response of delayed stormflow, including both peak intensity and cumulative effects. We appreciate your valuable feedback, which has helped us improve the clarity and precision of our manuscript. **(Page 2, Line 37)**

**Comment 5:**

Lines 58 & 59. Remark "a delayed peak only occurred when watershed storage reached a critical threshold of 113 mm" is given here as a result of general validity everywhere. I am afraid that it refers to a specific study area.

**Response 5:**

Thank you for your valuable comment. We agree that the statement regarding the threshold of 113 mm is specific to the study area of Martínez-Carreras et al. (2016). We have revised the text to clarify that this threshold value is based on their findings in that particular watershed, and we have added that the underlying processes behind this threshold are not yet fully understood. The revised sentence now reads:

"For instance, Martínez-Carreras et al. (2016) found that a delayed peak only occurred when watershed storage reached a critical threshold of 113 mm in their specific study area." **(Page 3, Line 61)**

**Comment 6:**

Line 112. Word "at" should be erased, shouldn't it?

**Response 6:**

Thank you for your helpful comment. You are correct that the word "at" should be erased. We have revised the sentence accordingly. **(Page 5, Line 115)**

**Comment 7:**

Line 113. Measurement units "m" should be added after "676".

**Response 7:**

Thank you for your careful review. We agree with your suggestion and have added the unit "m" after "676" in the revised manuscript. **(Page 5, Line 116)**

**Comment 8:**

Lines 127 & 132. The same measurement units for hydraulic conductivity should be used throughout the paper. I would prefer the use of SI units, so m/s.

**Response 8:**

Thank you for your valuable suggestion. We agree that consistent units should be used throughout the paper and that SI units (m/s) are preferable. We have converted all hydraulic conductivity values to m/s for consistency. The revised sentences now read:

"The soils in XEW are primarily brown earth and cinnamon soils, with depths up to 1.5 m and an average saturated hydraulic conductivity of $1.25 \times 10^{-5}$ m/s. [...] Slug tests estimated the saturated hydraulic conductivity of weathered granite to range from $6.02 \times 10^{-8}$ m/s to $1.34 \times 10^{-5}$ m/s." **(Page 6, Lines 130 & 135)**

**Comment 9:**

Line 139. Measurement units "m" should be added after "5".

**Response 9:**

Thank you for your careful review. We agree with your suggestion and have added the unit "m" after "5" in the revised manuscript **(Page 7, Line 143)**

**Comment 10:**

Lines 140 & 141. "2.3 Meteorological and streamflow 140 data collection" is the title of the next subsection.

**Response 10:**

Thank you for pointing this out. We acknowledge the formatting issue and have corrected it by ensuring proper spacing and separation between the sentence and the subsection title. **(Page 7, Line 145)**

**Comment 11:**

Line 142. Expression "spanning 2013-2023" should be rephrased, possibly as "spanning eleven years, from 2013 to 2023,".

Line 143. Measurement units "m" should be added after "700", "900", and "1000".

**Response 11:**

Thank you for your helpful suggestions. We have revised the sentence to improve clarity and consistency. The updated version now reads:

"Meteorological data spanning eleven years, from 2013 to 2023, were collected from four GRWS100 automatic weather stations (WS700, WS900, WS1000, and WS1100), positioned at elevations of 700m, 900m, 1000m, and 1100 m, respectively." **(Page 7, Lines 146 & 148)**

**Comment 12:**

Lines 145 & 155. Which kind of average? Space or time average? In the latter case, which time interval has been considered for the averaging?

**Response 12:**

Thank you for your insightful comment. The rainfall data were spatially averaged across six tipping-bucket rain gauges at each 10-minute interval. The revised sentences now read:

"Rainfall was recorded at 10-minute intervals using six tipping-bucket rain gauges near the weather stations, and the data were spatially averaged across the gauges for each time step for analysis." **(Page 7, Lines 149-150)**

The SWC data were also spatially averaged across multiple monitoring sites at each 10-minute interval. The revised sentences now read:

"Measurements were recorded every 10 minutes, and the arithmetic mean of SWC values across the monitoring sites was computed for each time step." **(Page 7, Line 160)**

**Comment 13:**

Lines 161 & 162. Sentence "Groundwater levels were normalized following the method described by Detty and McGuire 161 (2010)" repeats what was written at lines 158 & 159.

**Response 13:**

Thank you for your careful review. We acknowledge the redundancy and have removed the repeated sentence at lines 158 & 159 to improve clarity and conciseness. **(Page 8, Lines 165-166)**

**Comment 14:**

Lines 162 & 163. If I understood correctly, "the lowest recordable depth of the instrument" depends on the sensor position, so it is a factor related to the experimental setup, not a physical property of the subsurface.

**Response 14:**

Thank you for your insightful comment. We agree that the lowest recordable depth of the instrument is determined by sensor placement. However, the wells used in this study were drilled to sufficient depths to capture long-term groundwater fluctuations, ensuring that the observed groundwater variations reflect actual subsurface hydrological processes rather than limitations imposed by the experimental setup.

To improve clarity, we have revised the description of our groundwater level normalization approach as follows:

"GWLs (below the ground surface, hereinafter referred to as bgs) were observed in six boreholes distributed across the hillslopes. Hourly data were recorded using HOBO capacitance water level loggers. To facilitate comparisons across wells with varying absolute GWL ranges, we normalized the GWLs following the method described by Detty and McGuire (2010). Specifically, for each well, GWLs were normalized to their total observed range, assigning a value of 0 to the shallowest GWL and 1 to the deepest. The arithmetic means of these normalized values across all boreholes, referred to as the groundwater index ($I_G$), effectively represent the overall GWL dynamics in the watershed. Given that lower IG values indicate higher GWLs, and higher IG values correspond to deeper GWLs, figures presenting IG trends (e.g., Fig. 12 and Fig. A1) use an inverted vertical axis to align visually with hydrological intuition." **(Page 8, Lines 162-177)**

This revision clarifies that the normalization was applied to the entire range of observed GWLs, rather than being constrained by the lowest recordable depth of the instrument. The minimum observed GWL at each site is influenced by local geological and hydrological characteristics, such as bedrock depth and subsurface connectivity, making it a meaningful hydrological parameter rather than merely an experimental constraint. Similar normalization approaches have been widely used in previous studies (e.g., Detty & McGuire, 2010) to facilitate cross-site comparisons of

groundwater dynamics. We appreciate your feedback, which has helped us improve the clarity of our methodology.

**Comment 15:**

Lines 172 & 173. The definition of "event" should be better rephrased. In particular, the rainwater intensity is considered as the hourly averaged value or the value measured at the sampling period (10 minutes)?

**Response 15:**

Thank you for raising this important clarification. We have revised the definition to explicitly specify the temporal scale of rainfall intensity measurement. The modified text now reads:

"Rainfall events were identified using an intensity-based automatic algorithm (Tian et al., 2012) that defines an event as periods with hourly-averaged rainfall intensity exceeding 0.1 mm/h, separated by at least six consecutive hours with intensities below this threshold. Events with cumulative rainfall >5 mm were retained for analysis." **(Page 8, Lines 182-186)**

This revision aligns with the data processing methodology stated in Section 2.4:

"Rainfall, streamflow, and SWC data were aggregated to hourly intervals for temporal alignment with GWL measurements." **(Page 8, Line 178)**

**Comment 16:**

Line 177. ", HESS" should be erased.

**Response 16:**

Thank you for your suggestion. We agree with your comment and have removed "HESS" for accuracy. **(Page 9, Line 191)**

**Comment 17:**

Line 181. "&" should be substituted with "and" to be consistent throughout the whole paper.

**Response 17:**

Thank you for your careful review. We agree with your suggestion and have replaced "&" with "and" for consistency. **(Page 9, Line 195)**

**Comment 18:**

Lines 184 & 185. Erase these lines, they repeat lines 180 & 181.

**Response 18:**

Thank you for your careful review. We agree with your comment and have removed the redundant lines to avoid repetition. **(Page 9, Lines 194-195)**

**Comment 19:**

Lines 182 & 183, 186 & 187. These sentences basically repeat the same concepts and should be merged.

**Response 19:**

Thank you for your valuable comment. We acknowledge the redundancy in our original text and have merged the two sentences for conciseness. The revised sentence now reads:

"Throughout the manuscript, stormflow refers to the total discharge, while event stormflow volume ($q_s$) is calculated as the total discharge minus baseflow." **(Page 9, Lines 196-201)**

**Comment 20:**

Line 189. Is "was" correct? May be, "were"?

**Response 20:**

Thank you for pointing this out. We have corrected "was" to "were." **(Page 9, Line 203)**

**Comment 21:**

Line 191. Lowercase should be used for "the".

**Response 21:**

Thank you for pointing this out. We apologize for the oversight and have corrected "The" to lowercase. **(Page 9, Line 205)**

**Comment 22:**

Lines 253 & 254. A standard deviation of 0.01158, for 14 data, corresponds to a standard error on the average value of $0.0158/14^{1/2}=0.004$. Therefore, it would be better to write "a mean value of $0.197 \pm 0.004$ and…".

**Response 22:**

Thank you for your meticulous feedback. We sincerely appreciate your attention to statistical rigor and have revised the text accordingly. The corrected sentence now reads:

"The statistical analysis of the stable SWC revealed a mean value of $0.197 \pm 0.004$ m³/m³, with a confidence interval of 0.188–0.207 m³/m³." **(Page 12, Lines 268-269)**

**Key revisions:**

**Standard Error Clarification**:

Explicitly reported the standard error (SEM) as ±0.004, calculated via $SEM = \frac{0.0158}{\sqrt{14}} \approx 0.004$.

**Confidence Interval Correction**:

Recalculated the 95% confidence interval using the *t*-distribution ($t_{0.025,\ 13} = 2.16$):

$0.197 \pm 2.16 \times 0.0042 = [0.188, 0.207]$.

**Comment 23:**

Line 257. The caption should explain what is represented by the blue strip.

**Response 23:**

Thank you for your comment. The blue strip represents the 95% confidence interval of stable soil water content (SWC), with its upper and lower boundaries ranging from 0.189 to 0.205. We have revised the figure caption for clarity, and it now reads:

"Figure 5. SWC dynamics during different storm events. The blue strip indicates the 95% confidence interval of stable SWC." **(Page 13, Lines 273-274)**

**Comment 24:**

Line 288. Pronoun "that" should be added before "represents".

**Response 24:**

Thank you for your suggestion. We have revised the sentence by adding "which" before "represents" enhance grammatical clarity. **(Page 15, Line 305)**

The addition of the relative pronoun "which" ensures proper syntax for the non-restrictive clause describing $I_G$. This revision aligns with grammatical conventions while preserving the intended meaning. We appreciate your attention to detail in strengthening the manuscript's readability.

**Comment 25:**
Line 289. "Analysis revealed that" can be erased.
**Response 25:**
Thank you for your suggestion. We have removed "Analysis revealed that" to improve conciseness. **(Page 15, Line 306)**

**Comment 26:**
Line 335. "0.13-0.26" should be substituted as "from 0.13 to 0.26".
**Response 26:**
Thank you for your suggestion. We have revised "0.13-0.26" to "from 0.13 m³/m³ to 0.26 m³/m³" as recommended. **(Page 18, Line 352)**

**Comment 27:**
Lines 349 to 351. Expressions "0-2 h" and similar should be substituted, possibly as "less than 2 hours".
**Response 27:**
Thank you for your suggestion. We have replaced "0-2 h" with "less than 2 h," to improve readability. **(Page 18, Line 367)**

**Comment 28:**
Lines 364 & 365. Expression "a markedly faster rates (0.03 to 0.98/day, mean: 0.38/day) compared to the second peak 364 (0.01 to 0.31/day, mean: 0.07/day)" should be rephrased, possibly as "a markedly faster rate (from 0.03 d-1 to 0.98 d-1, with a mean of 0.38 d-1) than the second peak (from 0.01 d-1 to 0.31 d-1, with a mean of 0.07 d-1)."
**Response 28:**
Thank you for your valuable suggestion. We have revised the sentence for clarity and grammatical correctness. The updated version now reads:
"The first IG peak exhibited a markedly faster rate (ranging from 0.03 $d^{-1}$ to 0.98 $d^{-1}$, with a mean of 0.38 $d^{-1}$) than the second peak (ranging from 0.01 $d^{-1}$ to 0.31 $d^{-1}$, with a mean of 0.07 $d^{-1}$)." **(Page 19, Lines 382-384)**

**Comment 29:**
Figure 11. The notation of the measurement units of the principal vertical axis should be corrected (see previous comment).
**Response 29:**
Thank you for your comment. We have corrected the notation of the measurement units on the principal vertical axis of Figure 11, changing it to $d^{-1}$ for consistency and accuracy.

[Figure]

**Figure 11.** Growth rates of $I_G$ and the maximum $I_G$ value across storm events. $r_1$ and $r_2$ denote the ascent rates during the first and second peaks, respectively. **(Page 20, Line 390)**

**Comment 30:**
Line 379. "pace. pressure-driven" should be corrected. May be, "Pressure" should start with uppercase.

**Response 30:**
Thank you for your careful review. We have corrected the lowercase "pressure" to "Pressure" to ensure proper grammar and readability. **(Page 21, Line 399)**

**Comment 31:**
Lines 398 & 399. Expression "of 10-30 meters" should be substituted with "varying from 10 m to 30 m".

**Response 31:**
Thank you for your suggestion. We have " of 10-30 meters " with " varying from 10 m to 30 m" to improve readability. **(Page 21, Lines 418-419)**

**Comment 32:**
Figure 12. Figure caption must be rephrased in a more clear way. Why is this figure drawn with the IG axis increasing downwards? I found this choice confusing, when I was reading the comments about what happens for high and low values of $I_G$.

**Response 32:**
Thank you for your insightful comment. We acknowledge that the choice to plot $I_G$ with an inverted vertical axis may not have been immediately intuitive to all readers. However, this decision was made to align with the definition of the groundwater index ($I_G$) in our study.

As described in Section 2.4 (Methodology), $I_G$ is derived from normalized groundwater levels (GWLs) following the approach of Detty and McGuire (2010). Specifically, for each well, GWLs were normalized to their total observed range, where:

- A value of 0 represents the shallowest observed GWL (i.e., highest groundwater level).
- A value of 1 represents the deepest observed GWL (i.e., lowest groundwater level).

Since higher $I_G$ values correspond to lower groundwater tables and lower $I_G$ values indicate higher groundwater levels, we plotted $I_G$ on an inverted axis so that increasing $I_G$ values appear downward. This ensures that the figure visually aligns with hydrological intuition—higher groundwater levels (lower $I_G$) correspond to greater connectivity and shorter lag times, as discussed in the text.

To enhance clarity and avoid confusion, we will take the following actions:

1. Modify the figure caption to explicitly clarify the rationale for the inverted $I_G$ axis.
2. Add a brief explanation in Section 2.4 (where $I_G$ is defined) to reinforce this choice.

**Revised Figure Caption (Figure 12)**

"Figure 12. Correlation between peak $I_G$ and the time differences from peak GWL responses on HS1 and HS2 to HS3 ($\Delta t = t_S - t_{S3}$), where $t_{S1}$, $t_{S2}$, and $t_{S3}$ are the average lag times of peak GWLs on HS1, HS2, and HS3, respectively. Note that the $I_G$ axis is inverted: $I_G$ is a normalized groundwater index where lower values indicate higher GWLs, and higher values represent deeper GWLs." **(Page 23, Lines 442-443)**

**Additional Explanation in Section 2.4 (Methodology) (Revised Text)**

"To facilitate comparisons across wells with varying absolute GWL ranges, we normalized the GWLs following Detty and McGuire (2010). For each well, GWLs were normalized to their total observed range, assigning a value of 0 to the shallowest GWL (i.e., highest groundwater level) and 1 to the deepest GWL (i.e., lowest groundwater level). The arithmetic means of these normalized values across all boreholes, referred to as the groundwater index ($I_G$), effectively represent the overall GWL dynamics in the watershed. Given that lower $I_G$ values indicate higher groundwater levels, and higher $I_G$ values correspond to deeper groundwater tables, figures presenting $I_G$ trends (e.g., Fig. 12 and Fig. A1) use an inverted vertical axis to align visually with hydrological intuition." **(Page 8, Lines 164-177)**

**Comment 33:**

Lines 453 & 455. Isn't it better to give values in hours rather than in days?

**Response 33:**

Thank you for your suggestion. We have converted the rainfall duration and peak timing values from days to hours for better clarity and consistency. Accordingly, the relevant sentences now state: "Rainfall durations for the analyzed events ranged from 11h to 40 h. SWC, $I_G$, and delayed stormflow ($q_{2p}$) followed a clear sequence in their peak timings relative to rainfall onset. SWC responded rapidly, with its peak occurring 10 h to 50 h after rainfall began, usually coinciding with or slightly after rainfall cessation." **(Page 24, Lines 477 & 479)**

Additionally, we have updated the x-axis of Figure 13 from days to hours to maintain consistency.

[Figure]

**Figure 13.** Lag times of maximum SWC and GWL relative to rainfall onset. Each bar indicates the rise and peak times of the corresponding variable, with $t_{Rain}$ indicating rainfall duration. $SWC_{SP}$ and $I_{GP}$ represent the maximum SWC and $I_G$, respectively, while $q_{2p}$ denotes the delayed streamflow peak. **(Page 24, Line 464)**

The authors sincerely appreciate the editor's valuable feedback. We have thoroughly revised the manuscript to improve clarity, grammar, and fluency. The revised version has been submitted, and we look forward to your evaluation.

Yours

Zhen Cui, Fuqiang Tian

March 7, 2025

---

## Author Response (AR4)

**List of Changes**

1. Line 25 of Page 2: Replaced "while" with "and" for grammatical accuracy, in response to Comment 1.
2. Lines 208, 219, 223, 248, 249, 256–258, 274, 327, 331, 493, 497–499, 502, 507: Removed the unit notation "m³/m³" for SWC, in response to Comment 2.
3. Line 305 of Page 15: Revised "0.7–2.4 d" to "from 0.7 d to 2.4 d" and "0.2–3.6 d" to "from 0.2 d to 3.6 d" for improved readability, in response to Comment 3.

Note: The above changes are indicated using track changes in the marked-up revised manuscript.

**Response to Editors' Comments**

Dear Editor,

We sincerely appreciate your valuable comments and suggestions on our manuscript. In response, we have carefully reviewed your feedback and made the necessary revisions. Below, we provide a detailed point-by-point summary of the corrections made:

**Comment 1:**

Line 25. Correct grammar.

**Response 1:**

Thank you for your valuable feedback. To enhance clarity and grammatical precision, we have replaced "while" with "and" to more accurately reflect the relationship between delayed stormflow initiation and GWL fluctuations. The revised sentence is:

Delayed stormflow is initiated when SWC exceeds the soil's water storage capacity, and its timing and magnitude are further modulated by GWL fluctuations." **(Page 2, Line 25)**

**Comment 2:**

Lines 208, 219, 223, 248, 249, 256 to 258, 274, 327, 493, 497 to 499, 501, 507. SWC is dimensionless, it is useless to specify "m3/m3".

**Response 2:**

Thank you for your comment on the unit notation for SWC. While SWC is typically expressed as a dimensionless volume fraction, we initially used "m³/m³" for clarity. To align with standard hydrological conventions and avoid redundancy, we have removed the unit where unnecessary and revised the manuscript accordingly. **(Lines 208, 219, 223, 248, 249, 256 to 258, 274, 327, 331, 493, 497 to 499, 502, 507)**

**Comment 3:**

Line 326. Substitute "0.7–2.4 d" and "0.2–3.6 d" with "from 0.7 d to 2.4 d" and "from 0.2 d to 3.6 d", respectively.

**Response 3:**

We appreciate your suggestion to improve clarity by explicitly stating the range of lag times. We have revised the text accordingly, changing "0.7–2.4 d" to "from 0.7 d to 2.4 d" and "0.2–3.6 d" to "from 0.2 d to 3.6 d". Thank you for your valuable input. **(Page 15, Line 305)**

The authors sincerely appreciate the editor's valuable feedback. We have carefully revised the manuscript to enhance its clarity and rigor. The updated version has been submitted, and we look forward to your thoughtful evaluation.

Yours

Zhen Cui, Fuqiang Tian

March 9, 2025